# From Utterance to Vividity: Training Expressive Subtitle Translation LLM via Adaptive Local Preference Optimization

**Chaoqun Cui**[1,2], **Shijing Wang**[3], **Liangbin Huang**[3], **Qingqing Gu**[4], **Zhaolong Huang**[3], **Xiao Zeng**[3], **Wenji Mao**[1,2,*]
[1]MAIS, Institute of Automation, Chinese Academy of Sciences
[2]School of Artificial Intelligence, University of Chinese Academy of Sciences
[3]Hujing Digital Media & Entertainment Group
[4]Geely AI lab
`{cuichaoqun2025, wenji.mao}@ia.ac.cn`

## Abstract

The rapid development of Large Language Models (LLMs) has significantly enhanced the general capabilities of machine translation. However, as application scenarios become more complex, the limitations of LLMs in vertical domain translations are gradually becoming apparent. In this study, we focus on how to construct translation LLMs that meet the needs of domain customization. We take visual media subtitle translation as our topic and explore how to train expressive and vivid translation LLMs. We investigated the situations of subtitle translation and other domains of literal and liberal translation, verifying the reliability of LLM as reward model and evaluator for translation. Additionally, to train an expressive translation LLM, we constructed and released a multidirectional subtitle parallel corpus dataset and proposed the Adaptive Local Preference Optimization (ALPO) method to address fine-grained preference alignment. Experimental results demonstrate that ALPO achieves outstanding performance in multidimensional evaluation of translation quality.

## 1 Introduction

In recent years, the rapid advancement of Large Language Models (LLMs) has significantly enhanced the general capabilities of machine translation (Xu et al., 2024c; Enis & Hopkins, 2024; Feng et al., 2024). Representative LLMs such as GPT-4 (Achiam et al., 2023) and Qwen3 (Yang et al., 2025), trained on massive cross-lingual data, have demonstrated exceptional contextual comprehension and generation abilities in multilingual translation tasks. Their translation quality in general domains (e.g., news, daily conversations) has approached human-level performance (Xu et al., 2024a;b). However, as applications penetrate vertical domains, the limitations of LLMs in specialized translation scenarios have become increasingly apparent: critical issues include inadequate consistency in specialized terminology (An et al., 2024; Huang et al., 2024), deviations from industry-standard expressions (Ling et al., 2023; Pang et al., 2025), and weak style adaptability (Zhang et al., 2024). Consequently, LLM-based machine translation research has increasingly focused on addressing domain-specific customization requirements, such as developing legislation-focused LLMs that strictly adhere to clause formulation norms or medicine-oriented LLMs that precisely align with medical terminology.

In this study, we focus on visual media subtitle translation task, which aims to translate the lines in subtitles of visual media programs across genres such as movies, TV series, and documentaries from the source language into the target language. This task plays a crucial role in the localization of entertainment programs and in fostering global cultural dissemination and exchange (Federico et al., 2020; Wu et al., 2023; Mhaskar et al., 2024), yet it remains an underexplored subfield in machine translation. Similar to literary translation, subtitle translation requires a localized and stylistic **liberal translation** of program lines to convey the atmosphere, emotions, and tone of the original lines.

---

* Corresponding author.

However, while LLMs can achieve high translation accuracy, they tend to favor **literal translation** (as we quantitatively examined in Section 3.2). Therefore, exploring how to train a customized subtitle translation LLM with strong expressiveness and vividness is our primary challenge.

We employ LLM-as-a-Judge (using LLMs as evaluators) and preference optimization techniques to build a customized subtitle translation LLM. We conduct a quantitative investigation into the degree of liberal translation in human-translated texts across different domains and the liberal translation performance of various LLMs in subtitle translation. The findings reveal that: (1) Compared to the translation of serious texts requiring high accuracy (literal translation), such as legislation, news, and medicine, subtitle translation tends to favor more liberal translation; (2) In subtitle translation, the translations generated by chat LLMs tend to be more literal compared to those produced by reason LLMs and humans. We also experimentally validate the reliability of LLMs as reward models and evaluators for subtitle translation, as well as their alignment with human preferences. This enables automated construction of preference data for preference alignment training, thereby enhancing the expressiveness and vividness of subtitle translation LLMs. Additionally, for the task of translating short subtitle lines, which requires fine-grained local preference alignment, we propose a novel preference alignment strategy called **A**daptive **L**ocal **P**reference **O**ptimization (ALPO).

In summary, the main contributions of this study are as follows:

- We introduce the visual media subtitle translation task and investigate the extent of liberal translation and the reliability of LLM-as-a-Judge in this domain.
- We propose ALPO, a local preference optimization method, provide a formal explanation of its effectiveness, and use it to train a subtitle translation LLM with high vividness.
- We construct and release multi-directional subtitle parallel corpora based on an efficient bilingual subtitle alignment algorithm to support community research.
- Experimental results demonstrate that our 14B LLM, trained with ALPO, achieves significant improvements across multiple dimensions and outperforms SOTA LLMs.

## 2 RELATED WORK

### 2.1 LLM-AS-A-JUDGE

The capacity of LLMs to emulate human reasoning and evaluate specific inputs against a set of predefined rules has paved the way for "LLM-as-a-Judge" (Zheng et al., 2023a; Gu et al., 2024). Existing studies demonstrate that LLMs' scalability, adaptability, and cost-effectiveness render them particularly suitable for handling increasing volumes of assessment tasks that humans conventionally performed (Chen et al., 2024a; Li et al., 2024a; Wang et al., 2024c; Zhu et al., 2025). These capabilities are crucial for deploying LLMs flexibly across diverse evaluation scenarios and objectives, driving their rapid adoption in practical evaluation scenarios (Wang et al., 2024b; Chen et al., 2024b; Khattak et al., 2023).

Originally developed for language generation and comprehension tasks, LLMs have evolved significantly through advanced training methodologies such as Reinforcement Learning from Human Feedback (RLHF) (Ouyang et al., 2022), enhancing their alignment with human values and reasoning processes. This alignment has allowed LLMs to transition from generative tasks to evaluative roles. Fundamentally, LLM-as-a-Judge refers to the deployment of LLMs to assess objects, actions, or decisions according to predefined rules, criteria, or preferences (Kant, 1908; 1987). This framework encompasses a wide range of evaluative roles, including: Graders (Dong et al., 2023; Luong et al., 2024), Evaluators/Assessors (Li et al., 2024b; Zhang et al., 2023), Critics (Ke et al., 2024; Putta et al., 2024; Xiong et al., 2024), Verifiers (Shinn et al., 2023; Wang et al., 2024d), Examiners (Bai et al., 2023), Reward/Ranking Models (Luo et al., 2023; Sun et al., 2024; Yang et al., 2024b), etc.

### 2.2 LANGUAGE MODEL PREFERENCE OPTIMIZATION

Reinforcement learning provides an effective solution for aligning LLMs with human values and controlling text generation (Bender et al., 2021; Bommasani et al., 2021; Thoppilan et al., 2022; Taori et al., 2023; Chiang et al., 2023; Ji et al., 2023). To this end, RLHF framework based on human feedback reward models has been established (Ouyang et al., 2022; Christiano et al., 2017;

MacGlashan et al., 2017; Ziegler et al., 2019; Stiennon et al., 2020; Zheng et al., 2023b). However, despite its effectiveness, the complexity, instability, and hyperparameter sensitivity of RLHF remain insufficiently addressed (Andrychowicz et al., 2021; Engstrom et al., 2020). Recently proposed method named Direct Preference Optimization (DPO) (Rafailov et al., 2024) simplifies the RLHF framework by eliminating the need for explicit reward model construction or reinforcement learning procedures, thereby avoiding dependence on reward models. Several variants have subsequently emerged, such as SimPO, KTO, and IPO (Meng et al., 2024; Azar et al., 2024; Ethayarajh et al., 2024). Nevertheless, when applied to local preference alignment task, these methods still exhibit limitations, including coarse granularity and gradient dilution (see Appendix D for formal specification).

## 3 EMPIRICAL INVESTIGATION

### 3.1 LLM IS EXCELLENT TRANSLATION EVALUATOR

We aim to verify whether LLMs can achieve high consistency with human preferences in quality assessment across different dimensions of subtitle translation, thereby enabling LLMs to serve as both reward models and evaluators for subtitle translation model alignment training. We begin with a formal definition of LLM-as-Evaluator:

$$\mathcal{E} \leftarrow \pi_{\mathrm{e}}(\mathcal{C} \oplus s \oplus \{t\}), \tag{1}$$

where $\pi_{\mathrm{e}}$ is the LLM used for evaluation, $\mathcal{C}$ is the introduction and instruction in the prompt that guides the LLM's evaluation, $s$ is the line under evaluation, $\{t\}$ represents multiple translations of $s$, and $\mathcal{E}$ denotes the evaluation score given by $\pi_{\mathrm{e}}$.

We investigated the correlation between human evaluators and LLM evaluators in terms of vividness scores for 10 different translations of 500 lines from our self-constructed **Mu**ltilingual **S**ubtitle **C**orpus (MuSC) dataset in Appendix A. Each evaluator assigned a score from 0 to 100 for the 10 translations of each line under evaluation, and then we computed the average Spearman rank correlation $\rho$ across evaluators. We present the Spearman rank correlation results in Figure 1. The detailed experimental settings are provided in Appendix B.2.1. The translation directions include en⇒zh, en⇒de, zh⇒en, and zh⇒th, where Chinese and English, German, and Thai correspond to high-, medium-, and low-resource languages, respectively.

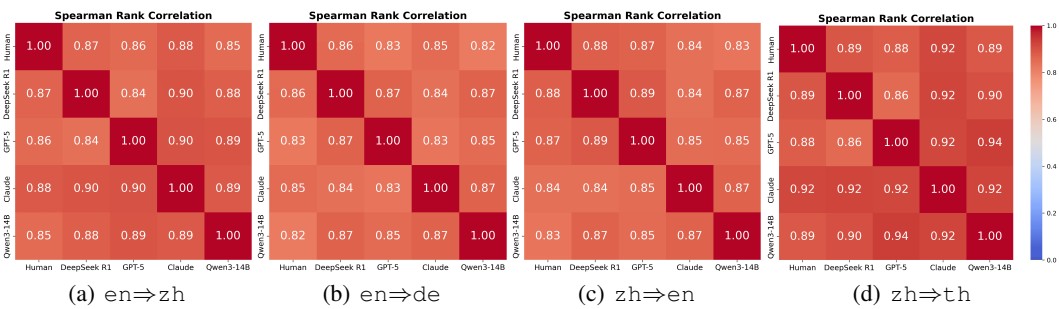

Figure 1: Investigation of the consistency of multiple evaluators with Spearman rank correlation $\rho$.

The results show that the LLM evaluator exhibits high consistency with human evaluators across multiple directions, indicating its ability to handle deviations in cultural and linguistic conventions across languages. Notably, even the Qwen3-14B model, with 14B parameters, achieves high agreement with both human evaluators and other SOTA LLMs across all directions ($\rho \geq 0.82$). This indicates that a 14B model, which incurs relatively low inference cost, can serve as an efficient reward model, forming the foundation of ALPO. Furthermore, Figure 2 presents the Bland-Altman plots of Qwen3-14B and human evaluators for identical line translations in en⇒zh and zh⇒th. The results show that the mean difference (MD) between the two is very low, indicating minimal systematic bias, and its limits of agreement (LoA) are within acceptable error margins for scoring on a scale of 100. This further confirms the consistency between the 14B model and human evaluators.

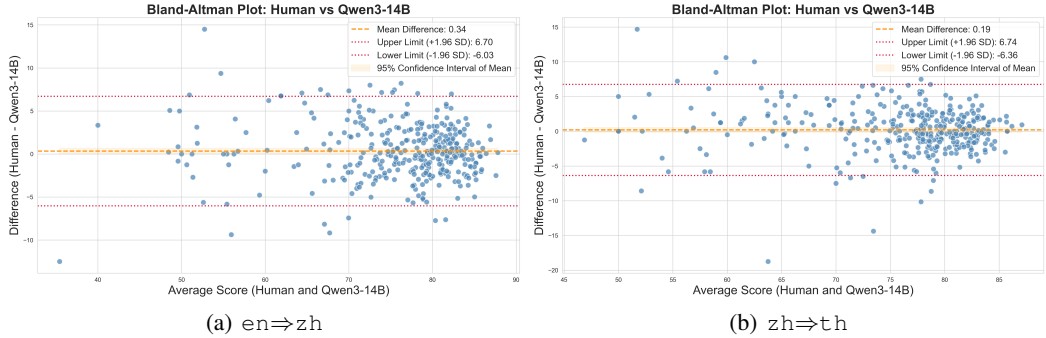

Figure 2: Bland-Altman plot comparing Qwen3-14B with human evaluator.

## 3.2 Parallel Corpora Are Actually Not Parallel

Literal translation focuses on translation accuracy, striving to preserve the original form, while liberal translation emphasizes conveying meaning and flexibly adjusting the expression (Baker & Saldanha, 2019; Munday et al., 2022). Based on our experience and observation, translations in different domains exhibit varying degrees of liberal translation: domains such as subtitle and literature translation often lean towards liberal translation, whereas legislation, medicine, and technical texts typically require highly accurate literal translation.

We designed relevant experiments to quantitatively validate this hypothesis. We investigated the back-translation consistency of parallel corpora from different domains. Specifically, for a particular translation direction, we utilized GPT-4o to directly translate the target language translation back to the source language, and then calculated the BLEU and ChrF++ scores between the back-translated text and the original text. Lower scores indicate a higher degree of liberal translation. Detailed experimental settings are provided in Appendix B.2.2.

The investigation results in Table 1 indicate that the back-translated texts in domains such as visual media, literature, and religion show lower similarity to the original texts compared to those in legislation, news, and medicine. This suggests a higher degree of liberal translation. For visual media subtitle translation, translators integrate video information into their translations, aiming to convey the atmosphere, emotions, and tone of the original text; this results in greater expressiveness and vividness, sometimes at the cost of sacrificing some degree of accuracy.

## 3.3 Chat LLMs Favor Literal, Reason LLMs Excel in Liberal

Generally, LLMs capable of effectively performing liberal translation tend to generate more expressive translations. We aim to verify the liberal translation capabilities of different LLMs. Our investigation spans multiple translation directions, where we sampled 2,000 lines from MuSC test set for each direction and prompted various LLMs to produce expressive and vivid translations (see the prompt in Appendix F.1). We then computed the pairwise BLEU similarity between human and multiple LLM-generated translations. The LLMs included are chat models such as GPT-4o (Achiam et al., 2023), Qwen-Max (Yang et al., 2024a), and Claude Opus 4.1 (Anthropic, 2025), as well as reasoning models like GPT-5 Thinking (OpenAI, 2025) and DeepSeek-R1 (Guo et al., 2025). Additionally, we incorporated our supervised fine-tuning (SFT) Qwen2.5-14B model trained on MuSC.

The results in Figure 3 indicate that translations generated by chat models exhibit higher similarity among themselves, suggesting a tendency toward more literal translation. In contrast, translations produced by reasoning models show lower similarity with other translations, demonstrating that these models, through thinking, better adhere to the instruction of liberal translation. This validates that LLM's inference-time scaling can effectively enhance translation performance. The human translation exhibits lower similarity with other translations, indicating its proficiency in liberal translation. We present demos of different translations in Appendix C.2 to visually demonstrate these findings.

Table 1: Domain liberal translation investigation. The highest and lowest are in blue and orange.

| Dataset | Domain | en⇒de | | en⇒fr | | en⇒es | |
|---|---|---|---|---|---|---|---|
| | | **BLEU** | **ChrF++** | **BLEU** | **ChrF++** | **BLEU** | **ChrF++** |
| OpenSubtitles | Visual Media | **15.00** | **37.08** | **17.84** | **37.80** | 21.60 | 43.78 |
| Books | Literature | 17.22 | 43.20 | 21.51 | 47.14 | **18.60** | **43.00** |
| bible-uedin | Religion | 22.55 | 49.57 | 22.07 | 48.24 | 22.31 | 51.03 |
| DGT | Legislation | 19.85 | 50.05 | 22.44 | 48.10 | 21.88 | 52.24 |
| JRC-Acquis | | **27.55** | **61.95** | **28.83** | **69.65** | 25.16 | 65.30 |
| News-Commentary | News | 25.00 | 55.23 | 24.87 | 54.37 | **35.90** | 65.84 |
| ECDC | Medicine | 23.18 | 60.77 | 23.86 | 62.97 | 27.20 | 65.39 |
| EMEA | | 25.20 | 59.36 | 24.73 | 61.65 | 31.46 | **69.15** |

| Dataset | Domain | de⇒en | | fr⇒en | | es⇒en | |
|---|---|---|---|---|---|---|---|
| | | **BLEU** | **ChrF++** | **BLEU** | **ChrF++** | **BLEU** | **ChrF++** |
| OpenSubtitles | Visual Media | 15.25 | **39.41** | **16.60** | **40.65** | 22.18 | 47.45 |
| Books | Literature | **14.50** | 40.50 | 20.12 | 48.29 | **15.84** | **43.99** |
| bible-uedin | Religion | 15.86 | 47.14 | 22.79 | 51.85 | 24.55 | 53.77 |
| DGT | Legislation | 19.70 | 47.77 | 23.31 | 48.91 | 22.87 | 52.49 |
| JRC-Acquis | | **27.54** | **63.43** | **31.33** | 64.15 | 27.68 | 64.46 |
| News-Commentary | News | 22.64 | 54.05 | 29.34 | 56.20 | **41.75** | 66.35 |
| ECDC | Medicine | 21.09 | 57.41 | 29.13 | **64.81** | 29.62 | 64.20 |
| EMEA | | 24.94 | 60.16 | 26.24 | 62.96 | 29.86 | **67.09** |

(a) en⇒zh     (b) en⇒de     (c) zh⇒en     (d) zh⇒th

Figure 3: Pairwise BLEU scores between translations.

## 4 METHOD

### 4.1 NOTATIONS

A bilingual dataset $\mathbb{D}$ for subtitle translation comprises collections of source language lines $\mathcal{L}_{\text{src}}$ and target language lines $\mathcal{L}_{\text{tgt}}$ from multiple programs of genres such as films or TV series, represented by $\mathbb{D} \equiv \{(\ell_{\text{src}}, \ell_{\text{tgt}}) \in \mathcal{L}_{\text{src}} \times \mathcal{L}_{\text{tgt}}\}$. The lines in multilingual subtitles of the same program are often not in one-to-one correspondence. To address this, we have developed a bilingual parallel corpus construction algorithm with $O(N)$ complexity; details can be found in Appendix A.2. We will utilize the dataset $\mathbb{D}$ to perform SFT and ALPO training on off-the-shelf LLMs. The input prompt $x$ for the LLM includes the context $\mathcal{C}$ (comprising introduction and instructions) and a set of $n$ source lines $\{s_i \mid i \in [n]\}$ that need to be translated, i.e., $x \equiv \mathcal{C} \oplus \{s_i\}$. The response $y$ from the LLM includes translations for each line $\{t_i \mid i \in [n]\}$, i.e., $y \equiv \{t_i\}$. See Appendix F.1 for detailed $x$ and $y$ format.

### 4.2 OVERALL FRAMEWORK

Obviously, constructing an expressive translation LLM is a preference optimization task. However, we cannot directly apply general methods (e.g., PPO (Ouyang et al., 2022) or DPO (Rafailov et al., 2024)), primarily because the translation of each line of subtitle depends on its context. The input $x$ of the SFT model $\pi_{\text{sft}}$ must include multiple lines of subtitle $\{s_i\}$, necessitating finer-grained alignment for each line in the response $y$. Consequently, outcome-supervised methods like PPO and DPO, which optimize the complete output of LLM, are inadequate for this task. We present experimental

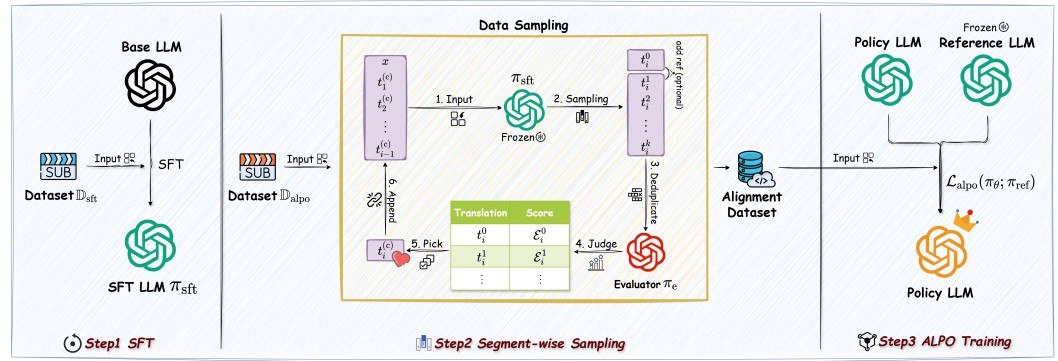

Figure 4: The overall framework of ALPO.

and theoretical validation of this in Appendix C.4 and Appendix D. To address this fine-grained preference optimization task, we propose the process-supervised ALPO method. ALPO introduces a novel paradigm that leverages a segment-wise sampling strategy and fine-grained alignment loss to train a high-quality subtitle translation LLM. The overall framework of ALPO is shown in Figure 4.

## 4.3 SAMPLING STRATEGY

We refer to a task requiring multi-segment local preference alignment of LLM response as a **local preference optimization** task. ALPO provides an effective paradigm for handling such tasks. We initially divide the parallel corpus $\mathbb{D}$ into a demonstration dataset $\mathbb{D}_{\text{sft}}$ and an alignment dataset $\mathbb{D}_{\text{alpo}}$ (approximately 8:2), using $\mathbb{D}_{\text{sft}}$ to train a SFT model $\pi_{\text{sft}}$ (see Appendix F.1 for input-output format).

In sampling phase, for a $x \in \mathbb{D}_{\text{alpo}}$, which contains $n$ lines $s_1, \ldots, s_n$. For each $s_i$, using $p_i = x, t_1^{(c)}, \ldots, t_{i-1}^{(c)}$ as prefix, $k$ translations are sampled ($k = 15$ in experiments), i.e., $\pi_{\text{sft}}(t_i^j \mid p_i)$, $j = 1, \ldots, k$, resulting in a candidate set $\{t_i^j \mid j = 1, \ldots, k\}$. Then, the candidate set is deduplicated; if human reference $t_i^0$ is available, it can be added to the candidate set, forming $\mathcal{T}_i = \{t_i^j \mid j = 0, 1, \ldots\}$. Subsequently, a Qwen3-14B model is deployed as an evaluator $\pi_e$ (or can be regarded as a reward model) to evaluate vividness of $\mathcal{T}_i$, obtaining the score sequence $\mathcal{E}_i$ (see Table 2 for demonstration). Based on $\mathcal{E}_i$, we select a superior translation $t_i^{(c)}$ for $s_i$ (randomly chosen from the top 3 scores) to serve as the prefix for the next sampling cycle. Ultimately, for a $x \in \mathbb{D}_{\text{alpo}}$, we obtain the sampling result $\mathcal{S}(x) \equiv \{(s_i, \mathcal{T}_i, \mathcal{E}_i) \mid i = 1, \ldots, n\}$. Algorithm 1 details the entire process.

Table 2: The zh⇒en evaluation demonstration of the evaluator $\pi_e$.

| Line | Evaluation of $\pi_e$ |
|---|---|
| 历史虽然会重演，但是人类是无法回到过去的。 | - |
| History repeats, but we can't go back to what was. | 70 |
| History might replay, but mankind cannot go back in time. | 85 |
| The wheel of history may turn full circle, but the door to the past stays forever locked. | 88 |
| Although history may repeat itself, humans cannot return to the past. | 82 |
| History often echoes, yet there's no way for us to turn back the clock. | 92 |

## 4.4 ADAPTIVE ALIGNMENT LOSS

The reward signal acts on segments of model output (process-supervised) rather than the complete response (outcome-supervised), which constitutes the main challenge for vividness enhancement. ALPO enables process-supervised alignment across multiple segments in translation LLM's response $y$. Specifically, for all lines $\{s_i \mid i = 1, \ldots, n\}$ in an $x \in \mathbb{D}_{\text{alpo}}$, we assign each $s_i$ an adaptive weight $w(s_i)$, defined as the product of a gating function $\mathbf{1}(s_i)$ and an importance score $\delta(s_i)$:

$$w(s_i) = \mathbf{1}(s_i) \cdot \delta(s_i), \tag{2}$$

where $\mathbf{1}(s_i)$ acts as a gate to determine whether $s_i$ participates in optimization. When the sampled translations lack diversity or clear quality distinction, i.e., $|\mathcal{T}_i| \leq 3$ or $\max(\mathcal{E}_i) - \min(\mathcal{E}_i) \leq 5$, we

---

**Algorithm 1** ALPO Sampling Strategy.

---
**Input:** SFT model $\pi_{\text{sft}}$, evaluation LLM $\pi_{\text{e}}$, alignment dataset $\mathbb{D}_{\text{alpo}}$, sample number $k$.
**Output:** sampled segment-level candidate set $\mathcal{S}(x)$.
 1: **for** any $x \in \mathbb{D}_{\text{alpo}}$ **do**      // Iterate through the alignment dataset $\mathbb{D}_{\text{alpo}}$.
 2:     **for** $i = 1$ to $n$ **do**      // Iterate through the subtitle lines in $x$.
 3:         **for** $j = 1$ to $k$ **do**      // Sample multiple translation candidates.
 4:             Sample $\pi_{\text{sft}}(t_i^j \mid x, t_1^{(\text{c})}, \ldots, t_{i-1}^{(\text{c})})$.
 5:         **end for**
 6:         Deduplicate the candidate set $\{t_i^j \mid j = 1, \ldots, k\}$.
 7:         (Optional) Add human reference $t_i^0$ to the candidate set, get $\mathcal{T}_i = \{t_i^j \mid j = 0, 1, \ldots \}$.
 8:         Evaluate $\mathcal{T}_i$ by $\pi_{\text{e}}$, get score sequence $\mathcal{E}_i$.
 9:         Select a superior translation $t_i^{(\text{c})}$ (random from top 3) as the prefix of next sampling cycle.
10:     **end for**
11: **end for**
12: **return** $\mathcal{S}(x) \equiv \{(s_i, \mathcal{T}_i, \mathcal{E}_i) \mid i = 1, 2, \ldots, n\}$.

---

set $\mathbf{1}(s_i) = 0$; otherwise, $\mathbf{1}(s_i) = 1$. The importance score $\delta(s_i)$ depends on the diversity of sampled translations. Since lines with richer translations provide more potential for vividness enhancement, we assign higher $\delta(s_i)$ to lines with a larger number of distinct sampled translations after deduplication:

$$\delta(s_i) = \frac{|\mathcal{T}_i|}{\sum_{j=1}^n |\mathcal{T}_j|}. \tag{3}$$

We then apply a Bradley–Terry (DPO-style) preference optimization loss to achieve fine-grained preference alignment of the translation model:

$$\mathcal{L}_{\text{alpo}}(\pi_\theta; \pi_{\text{ref}}) = -\mathbb{E}_{(x,\mathcal{S}(x))\sim\mathbb{D}_{\text{alpo}}} \left[ \sum_{i=1}^n w(s_i) \cdot \log \sigma \left( \beta_i \log \frac{\pi_\theta(t_i^{(\text{c})} \mid p_i)}{\pi_{\text{ref}}(t_i^{(\text{c})} \mid p_i)} - \beta_i \log \frac{\pi_\theta(t_i^{(\text{r})} \mid p_i)}{\pi_{\text{ref}}(t_i^{(\text{r})} \mid p_i)} \right) \right]. \tag{4}$$

where $\pi_\theta$ is the policy model, and $\pi_{\text{ref}}$ is the reference model. The chosen translation $t_i^{(\text{c})}$ is randomly sampled from the top 3 scored translations in $\mathcal{E}_i$, while the rejected translation $t_i^{(\text{r})}$ is selected as the third-lowest scored candidate (excluding the lowest one to avoid overly trivial contrastive pairs). $\beta_i$ is a hyperparameter controlling sensitivity to reward differences. Since the reward gap between $t_i^{(\text{c})}$ and $t_i^{(\text{r})}$ also reflects the diversity of translations, we set $\beta_i$ dynamically as follows:

$$\beta_i = \frac{r(s_i, t_i^{(\text{c})}) - r(s_i, t_i^{(\text{r})})}{\max\{r(s_j, t_j^{(\text{c})}) - r(s_j, t_j^{(\text{r})}) \mid j \in [n]\}}, \tag{5}$$

where $r(s_i, t_i^{(\text{c})})$ denotes the score of $t_i^{(\text{c})}$ in $\mathcal{E}_i$. Moreover, instead of directly setting the prefix in $\mathcal{L}_{\text{alpo}}$ as $p_i = x, t_1^{(\text{c})}, \ldots, t_{i-1}^{(\text{c})}$, we adopt a scheduled prefix mixing strategy to mitigate exposure bias when non-chosen translations are generated during inference. Specifically, with probability $\lambda$, $t_i^{(\text{c})}$ is appended to the prefix, and with probability $1 - \lambda$, a translation sampled from $\mathcal{T}_i$ is appended (in experiments, $\lambda$ is increased from 0.2 to 0.6 as training progresses), i.e.,

$$p_i = x, \hat{t}_1, \ldots, \hat{t}_{i-1}, \quad \hat{t}_j \leftarrow \text{Mix}(t_j^{(\text{c})}, t_j \sim \mathcal{T}, \lambda). \tag{6}$$

## 5 EXPERIMENTS

### 5.1 EXPERIMENTAL SETTINGS

We run experiments using our customized MuSC dataset, which comprises subtitle corpora from the online video platform Youku for multiple directions, including en⇒de, en⇒fr, en⇒zh, ko⇒zh, zh⇒en, and zh⇒th. Each direction includes 100–200 programs across various genres, with 10% of the programs reserved as the test set. See Appendix A for details of MuSC.

We compare our model with the following baselines:

- **VideoDubber** (Wu et al., 2023) constructs a length-controlled translation model (the work most closely related to our newly proposed subtitle translation task).
- **NLLB-3.3B** (Costa-jussà et al., 2022) is a translation model of Meta AI (>200 languages).
- **MADLAD-10B** (Kudugunta et al., 2024) is a translation model of Google (>450 languages).
- **Google Translate** (Google LLC, 2023) is a multilingual translation service of Google.
- **GPT-4o**, **Qwen-Max**, and **DeepSeek-V3.1** (Liu et al., 2024) are current SOTA chat models.
- **DeepSeek-R1** and **GPT-5 Thinking** are SOTA reasoning models. We use the prompt in Appendix F.1 to drive these LLMs for subtitle translation.
- **Qwen2.5-14B** serves as the backbone for ALPO.

Details of the experimental setup are provided in Appendix B. The source code for ALPO and MuSC dataset are available at `https://github.com/CcQunResearch/ALPO`.

## 5.2 TRANSLATION QUALITY EVALUATION

Unlike the strict requirements for accuracy in the translation of legal texts and technical documents, subtitle translation focuses more on the localized representation of the translation (Pérez-González, 2019; Bajčić & Golenko, 2024). Specifically, subtitle translation demands not only the conveyance of the original meaning but also expects to reflect the emotions and tone of the original subtitle, aiming for translations that are more fluid, dynamic, and expressive. Given that we have verified the reliability of LLMs as evaluators in Section 3.1, we developed a multidimensional quality evaluation system for subtitle translation based on LLM-as-a-Judge scoring. We primarily assess three dimensions of subtitle translation: (1) *Accuracy*: whether the translation accurately conveys the original meaning of the subtitle; (2) *Naturalness*: whether the expression in the translation is natural and fluent, aligning with the grammatical structure and lexical conventions of the target language; (3) *Vividness*: whether the translation is expressive and successfully conveys the emotions and atmosphere of the original subtitle. We employ DeepSeek-V3.1, Claude Sonnet 4, and GPT-5 Instant as evaluation models to score subtitle segments in the test set along three dimensions, assigning a score between 0 and 100 for each. Then, we show the average score across all models in Table 3.

Table 3: Multidimensional quality evaluation. The 1st and 2nd best results are denoted as blue and orange. ST for supervised training. ICL for in-context learning.

| Models | Training | en⇒de | | | en⇒fr | | | en⇒zh | | |
|---|---|---|---|---|---|---|---|---|---|---|
| | | Accuracy | Naturalness | Vividness | Accuracy | Naturalness | Vividness | Accuracy | Naturalness | Vividness |
| Gold Reference | Human | 84.8 | 83.8 | **73.1** | 83.5 | 85.0 | 74.8 | 83.6 | 82.6 | **71.5** |
| VideoDubber | | 41.5 | 35.5 | 41.0 | 48.2 | 47.3 | 48.5 | 46.9 | 51.9 | 49.7 |
| NLLB-3.3B | ST | 75.8 | 70.1 | 59.2 | 76.9 | 74.6 | 61.8 | 61.4 | 54.0 | 43.7 |
| MADLAD-10B | | 73.2 | 65.6 | 54.5 | 73.6 | 67.8 | 57.4 | 59.7 | 55.5 | 46.3 |
| Google Translate | | 89.9 | 80.6 | 62.6 | 91.9 | 84.8 | 64.3 | 84.2 | 79.7 | 54.4 |
| GPT-4o | | 94.1 | 86.7 | 66.9 | 93.2 | 88.8 | 69.5 | 89.3 | 82.3 | 59.8 |
| Qwen-Max | ICL (C) | **95.5** | **89.2** | 68.9 | **94.1** | 89.9 | 71.6 | **91.9** | 84.4 | 61.3 |
| DeepSeek-V3.1 | | 94.8 | **89.2** | 67.4 | **94.9** | **90.7** | 72.3 | 91.2 | 85.3 | 63.5 |
| DeepSeek-R1 | ICL (R) | 92.8 | 87.4 | 70.0 | 93.6 | 90.0 | 73.4 | 90.5 | **85.7** | 70.8 |
| GPT-5 | | 93.6 | 88.6 | 72.7 | 92.2 | **90.4** | **75.8** | **92.4** | **87.0** | 71.1 |
| Qwen2.5-14B | SFT | 87.5 | 83.6 | 64.4 | 86.2 | 85.4 | 67.7 | 86.4 | 82.0 | 59.1 |
| Qwen2.5-14B | **ALPO** | **95.4** | 88.4 | **74.8** | 94.1 | 89.2 | **78.8** | 90.6 | 84.3 | **76.6** |

| Models | Training | ko⇒zh | | | zh⇒en | | | zh⇒th | | |
|---|---|---|---|---|---|---|---|---|---|---|
| | | Accuracy | Naturalness | Vividness | Accuracy | Naturalness | Vividness | Accuracy | Naturalness | Vividness |
| Gold Reference | Human | 78.0 | 77.8 | **65.8** | 83.0 | 80.3 | 73.3 | 76.6 | 75.1 | 66.3 |
| VideoDubber | | 39.6 | 45.2 | 48.2 | 53.6 | 54.8 | 50.1 | 34.1 | 34.9 | 41.5 |
| NLLB-3.3B | ST | 33.1 | 26.1 | 25.4 | 29.1 | 21.7 | 20.8 | 42.6 | 33.9 | 40.5 |
| MADLAD-10B | | 44.9 | 42.9 | 46.7 | 45.1 | 38.9 | 37.6 | 47.9 | 50.8 | 51.0 |
| Google Translate | | 54.9 | 52.8 | 52.0 | 79.8 | 66.3 | 50.2 | 55.2 | 56.2 | 54.5 |
| GPT-4o | | 80.0 | 79.9 | 58.1 | 88.5 | 83.0 | 64.6 | 88.0 | 84.4 | 67.9 |
| Qwen-Max | ICL (C) | 83.7 | 82.5 | 61.8 | **90.0** | 85.0 | 66.8 | **91.3** | **85.8** | 69.1 |
| DeepSeek-V3.1 | | 83.1 | 82.2 | 57.2 | **89.5** | 84.1 | 63.0 | 89.9 | 84.6 | 67.1 |
| DeepSeek-R1 | ICL (R) | 79.8 | 81.6 | 65.6 | 88.5 | 85.6 | 73.5 | 87.6 | 84.0 | 71.0 |
| GPT-5 | | **84.5** | **82.6** | 65.0 | 89.1 | **86.1** | **75.2** | 88.7 | 83.9 | **73.0** |
| Qwen2.5-14B | SFT | 80.9 | 76.1 | 53.9 | 85.2 | 80.1 | 54.8 | 87.3 | 82.6 | 66.0 |
| Qwen2.5-14B | **ALPO** | **84.3** | **83.3** | **70.5** | 88.3 | **86.8** | **81.7** | **91.9** | **84.7** | **74.2** |

The results indicate that ALPO and cutting-edge LLMs like GPT-4o significantly outperform traditional translation models like MADLAD, and surpass human translation in accuracy and naturalness. However, human translation excelled in vividness, possibly suggesting that human translations often incorporate more liberal translation by integrating video content. Reason models achieved a better vividness score compared to chat models, although its accuracy and naturalness were slightly lower

in some aspects. This aligns with the conclusions in Section 3.3. Models trained with ALPO demonstrated marked improvement not only in vividness but also in accuracy and naturalness compared to the SFT model. ALPO attained the highest vividness scores across all directions and occasionally surpassed other cutting-edge LLMs in other dimensions, particularly excelling in relatively low-resource language directions such as ko⇒zh and zh⇒th. For the complete results of quality assessment and the prompt design of the LLM evaluator, please refer to Appendix C.1 and Appendix F.3.

## 5.3 HUMAN EVALUATION OF TRANSLATION QUALITY

We present the human evaluation results of the translation quality of ALPO for en⇒zh and zh⇒th in Table 4. Due to varying subjective preferences among evaluators, we do not adopt a scoring system. Instead, we perform pairwise comparisons of different translations to assess the win rate metric. We evaluate the ALPO model alongside four baselines across dimensions of accuracy, naturalness, and vividness, and conduct a comprehensive assessment. The human evaluation results are consistent with those shown in Table 3, which supports the reliability of our multidimensional evaluation system based on LLM-as-a-Judge. Specific experimental settings can be found in Appendix B.3.

Table 4: Human translation quality evaluation, reporting win rate (win:tie:loss). The winning and losing contrasts are marked in blue and orange, respectively.

| Challenger | Competitors | en⇒zh | | | | zh⇒th | | | |
|---|---|---|---|---|---|---|---|---|---|
| | | Accuracy | Naturalness | Vividness | Comprehensive | Accuracy | Naturalness | Vividness | Comprehensive |
| | Gold Reference | 29:49:22 | 28:50:22 | 32:42:26 | 31:46:23 | 28:48:24 | 26:52:22 | 32:39:29 | 29:49:22 |
| ALPO | SFT Model $\pi_{sft}$ | 26:50:24 | 31:48:21 | 38:41:21 | 37:43:20 | 26:55:19 | 27:51:22 | 39:42:19 | 30:50:20 |
| (Qwen2.5-14B) | GPT-4o | 22:54:24 | 20:57:23 | 29:51:20 | 26:54:23 | 23:57:20 | 23:51:26 | 36:37:27 | 30:45:25 |
| | DeepSeek-R1 | 22:55:23 | 19:57:24 | 22:58:20 | 20:59:21 | 23:55:22 | 18:62:20 | 28:50:22 | 27:47:24 |

## 5.4 ABLATION STUDY

### 5.4.1 ADAPTIVE STRATEGY

We validate the impact of several adaptive strategies in ALPO on model performance in Table 5. Results show that the gating function $\mathbf{1}(s_i)$ has the greatest effect, indicating that the gating mechanism effectively reduces noise from low-diversity irrelevant lines (e.g., simple lines such as "Good morning."). Moreover, ablating the importance score $\delta(s_i)$, dynamic $\beta$, and scheduled prefix mixing also leads to varying degrees of performance drop, highlighting the necessity of fine-grained control in training.

Table 5: Impact of adaptive strategies.

| | en⇒zh | | | zh⇒th | | |
|---|---|---|---|---|---|---|
| | Accuracy | Naturalness | Vividness | Accuracy | Naturalness | Vividness |
| SFT | 86.5 | 82.1 | 59.2 | 87.2 | 82.4 | 66.0 |
| ALPO | 90.6 | 84.2 | 76.6 | 91.9 | 84.7 | 74.1 |
| w/o $w(s_i)$ | 88.1 | 83.2 | 67.4(↓9.2) | 89.2 | 83.4 | 70.3(↓3.8) |
| w/o $\mathbf{1}(s_i)$ | 89.1 | 83.4 | 70.2(↓6.4) | 89.9 | 83.9 | 71.2(↓2.9) |
| w/o $\delta(s_i)$ | 89.4 | 83.5 | 72.4(↓4.2) | 90.2 | 83.1 | 71.9(↓2.2) |
| fixed $\beta_i$ ($\beta = 0.5$) | 90.0 | 83.4 | 74.7(↓1.9) | 89.8 | 83.4 | 72.1(↓2.0) |
| w/o prefix mixing | 89.5 | 83.7 | 73.4(↓3.2) | 88.8 | 83.7 | 71.6(↓2.5) |

### 5.4.2 BACKBONE MODELS

We compare the performance of different backbone models in Table 6. The results demonstrate that ALPO enables all backbone models to achieve significant performance improvements compared to the SFT model (Qwen2.5-14B). Notably, the larger Qwen2.5-14B model with more parameters delivers superior and stable outcomes. LLaMA-3.1-8B exhibits underperformance in Chinese-related tasks, resulting in inferior results compared to the other two models in both en⇒zh and zh⇒en translation directions. Nevertheless, it still achieves measurable improvements over the SFT baseline.

Table 6: Impact of backbone models. The 1st and 2nd best results are marked as **blue** and **orange**.

| Method | Backbone | en⇒de | | | en⇒zh | | | zh⇒en | | |
|---|---|---|---|---|---|---|---|---|---|---|
| | | Accuracy | Naturalness | Vividness | Accuracy | Naturalness | Vividness | Accuracy | Naturalness | Vividness |
| SFT | - | *87.7* | *83.4* | *64.4* | *86.5* | *82.1* | *59.2* | *85.2* | *80.1* | *54.9* |
| ALPO | LLaMA-3.1-8B (Dubey et al., 2024) | 94.7 | 87.2 | 73.9 | 88.0 | 83.2 | 72.3 | 87.2 | 85.3 | 77.6 |
| | GLM4-9B (GLM et al., 2024) | 93.2 | 86.1 | 73.2 | 88.2 | 84.4 | 74.4 | 89.2 | 85.7 | 78.2 |
| | Qwen2.5-14B (Yang et al., 2024a) | 95.2 | 88.3 | 74.8 | 90.6 | 84.2 | 76.6 | 88.3 | 86.8 | 81.6 |

## 5.5 ADDITIONAL EXPERIMENTS

We perform the following extended experiments in the appendix: 1) In Appendix C.1, we present full results of quality evaluation with further analysis. 2) In Appendix C.2, we show demonstration examples of ALPO translations. 3) In Appendix C.3, We conduct ablation studies for other factors like samping and model size. 4) In Appendix C.4, we perform performance evaluation comparing DPO and PPO. 5) In Appendix C.5, we validate the effectiveness of ALPO in other application task.

## 6 CONCLUSION

In this study, we investigate literal and liberal translation in subtitle translation as well as other domains, and validate the reliability of large language models as evaluators and reward models for translation quality. Building upon this, we propose the Adaptive Local Preference Optimization (ALPO) method for training expressive and vivid subtitle translation. Our experiments, conducted under the established translation quality evaluation framework, validate the effectiveness of ALPO.

## ETHICS STATEMENT

This study utilizes subtitle data obtained exclusively from the online video platform Youku. All data employed in this study were acquired with proper authorization and comply with the terms and conditions set forth by Youku. The data were used solely for academic purposes and were handled in a manner that ensures compliance with relevant ethical and legal standards.

No personal or sensitive information was collected or processed in this research. All analyses were performed on accessible content provided by Youku, and no attempts were made to infer or reveal any private or identifying information about individuals or entities featured in the materials.

We affirm that this study adheres to the principles of responsible and ethical research and complies with all applicable institutional and regulatory guidelines.

## REPRODUCIBILITY STATEMENT

We release all code, training scripts, and the MuSC dataset used in this study to facilitate reproducibility and further research. Detailed experimental settings, model configurations, and evaluation protocols are provided in the main text and appendices. The source code and dataset are publicly accessible at `https://github.com/CcQunResearch/ALPO`.

## ACKNOWLEDGMENTS

This work is supported in part by the National Natural Science Foundation of China under Grant #72293575, and the Joint Research Project on the Integration of Culture, Science and Technology between Chinese Academy of Sciences and Hunan Province #2024JK4003.

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

## APPENDIX CONTENTS

| Appendix Sections | Contents |
| --- | --- |
| Appendix A | Data Source and Dataset Construction |
| Appendix B | Additional Details of Experimental Settings |
| Appendix C | Extended Evaluation Experiments for ALPO |
| Appendix D | Theory: Adaptive Local Preference Optimization |
| Appendix E | Further Discussions on ALPO |
| Appendix F | Prompts and Instructions in ALPO |

## A DATASET CONSTRUCTION

In this section, we will introduce the details related to the datasets used for model training.

### A.1 DATA SOURCES

In this study, we utilize the MuSC dataset from the online video platform Youku for experiments, which includes source language subtitles and multilingual translated subtitles of the platform's programs. Table 7 presents samples of English and Chinese subtitles from the MuSC dataset (in the ass file format), where the "Start" and "End" columns indicate the start and end times of the subtitle within the program. The translation directions covered in MuSC include en⇒de, en⇒fr, en⇒zh, ko⇒zh, zh⇒en, and zh⇒th. These translation directions encompass both cross-language family and cross-language branch scenarios. Subtitle translation, as a data-rich task, makes it easy for leading online video platforms to gather a substantial amount of multilingual subtitle corpus. The MuSC dataset includes over 100 programs in each direction, spanning genres such as film, TV series, documentary, and animation over multiple years. Statistics for MuSC dataset are provided in Table 8.

Unlike the translation of serious texts in fields such as legislation and medicine, which require a high degree of accuracy (AbuSa'aleek, 2016; Rashed Alkatheery, 2023; Bajčić & Golenko, 2024), subtitle

text in visual media programs is deeply bound to video and audio modal information, allowing for some tolerance of accuracy loss in translation. However, it is crucial that the translated subtitle accurately conveys the emotions, tone, atmosphere, and cultural context of the original text. Therefore, human translation does not simply involve direct translation of the original subtitle but rather includes polishing, rewriting, and adaptation to meet the localization needs of the target audience (Bassnett, 2013; Pérez-González, 2019). As a result, the gold reference translations produced by human translators are refined versions of the original text, implying that the parallel corpus used for training is not truly "parallel" (the conclusion addressed in Section 3.2), and there is information asymmetry between the original text and its translation (see Table 7 for an example). Consequently, while constructing the training set, each direction's program is a native-language program.

Table 7: Examples of multilingual subtitles.

| Start | End | Text |
|---|---|---|
| 0:17:33.25 | 0:17:34.50 | Please, let me speak! |
| 0:17:39.54 | 0:17:42.83 | As a lower house, my voice doesn't carry much weight here. |
| 0:17:42.91 | 0:17:46.45 | But as a mother, I have a voice that matters deeply. |
| 0:17:47.50 | 0:17:50.25 | My son isn't in his right mind. |
| 0:17:51.04 | 0:17:53.95 | His entire life, he's chased an impossible dream. |
| 0:17:54.91 | 0:17:59.00 | What he did was, uh, foolish and unwise. |
| 0:17:59.83 | 0:18:04.20 | But he has a good heart. Please, let him come home. |
| 0:18:04.29 | 0:18:08.66 | A crime like this can't be overlooked. The boy must be punished. |
| 0:18:08.75 | 0:18:12.16 | A violation of the Ethos calls for banishment, |
| 0:18:12.25 | 0:18:16.25 | but I can sympathize with a young man's dream to change the world. |
| 0:18:16.83 | 0:18:20.58 | Perhaps in this matter, a lesser sentence may suffice. |
| 0:17:42.91 | 0:17:46.50 | 我以母亲的身份恳请在座的各位议员们听一听 |
| 0:17:47.00 | 0:17:50.25 | 我这儿子确实犯了不可饶恕的错误 |
| 0:17:51.04 | 0:17:53.95 | 他这辈子 都在追逐一个不可能实现的梦想 |
| 0:17:54.91 | 0:17:59.33 | 他的所作所为是很愚蠢 很不明智的 |
| 0:17:59.83 | 0:18:04.25 | 但他并没有坏心 求你们 放他回家吧 |
| 0:18:04.33 | 0:18:06.70 | 这样严重的罪行不能轻易放过 |
| 0:18:06.79 | 0:18:08.66 | 这小子必须受到惩罚 |
| 0:18:08.75 | 0:18:12.20 | 违反社会共识的人确实应该遭到驱逐 |
| 0:18:12.29 | 0:18:16.25 | 但我也能体会一个年轻人梦想改变世界的雄心 |
| 0:18:16.33 | 0:18:20.58 | 就这个案子来说 还是酌情予以轻判为好 |

Table 8: Statistics of the datasets. FM, TV, DO, and AN are abbreviations for film, TV series, documentary, and animation, respectively.

| Statistic | en⇒de | en⇒fr | en⇒zh | ko⇒zh | zh⇒en | zh⇒th |
|---|---|---|---|---|---|---|
| cross language family | ✗ | ✗ | ✓ | ✓ | ✓ | ✓ |
| cross language branch | ✗ | ✓ | ✓ | ✓ | ✓ | ✓ |
| period | 2013-2024 | 2015-2024 | 2020-2024 | 2017-2024 | 2021-2024 | 2021-2024 |
| # programs | 210 | 207 | 161 | 139 | 225 | 218 |
| program types | FM,TV,AN | FM,TV,AN | FM,TV,AN | TV | FM,TV,DO | FM,TV,DO |
| # lines | 1.87M | 1.72M | 1.02M | 1.54M | 1.26M | 1.21M |
| total length (h) | 1156 | 1033 | 852 | 802 | 501 | 479 |
| # avg source token | 7.82 | 7.67 | 7.77 | 9.87 | 6.19 | 6.08 |
| # avg target token | 10.83 | 10.64 | 6.41 | 6.51 | 8.17 | 21.11 |

## A.2 BILINGUAL PARALLEL CORPUS CONSTRUCTION

Typically, the multilingual subtitles for the same program do not correspond line by line. This is mainly due to linguistic differences, which may lead to the merger or splitting of original line during human translation. Additionally, subtitle files may contain noise such as scene titles and annotations. To automate the line-by-line alignment of multilingual subtitles, we designed a subtitle alignment algorithm that utilizes subtitle timing.

For the source language and target language line sets $\mathcal{L}_{src}$ and $\mathcal{L}_{tgt}$, we first determine the row margin $M = abs(|\mathcal{L}_{src}| - |\mathcal{L}_{tgt}|)$. Then, for each line $\ell_{src}$ in $\mathcal{L}_{src}$, we search within a window of size 2M around the corresponding index in $\mathcal{L}_{tgt}$ for a line whose start time differs from that of $\ell_{src}$ by within 0.7 seconds to serve as $\ell_{tgt}$. The reason for choosing 0.7 seconds as the threshold is that the shortest sentence in various languages (such as "Hello") typically has a duration of about 0.7 seconds. Through

this process, we can automatically gather a training dataset $\mathbb{D} \equiv \{(\ell_{\text{src}}, \ell_{\text{tgt}}) \in \mathcal{L}_{\text{src}} \times \mathcal{L}_{\text{tgt}}\}$ from bilingual subtitles in programs. The complexity of this process is $O(N)$.

## B  ADDITIONAL DETAILS OF EXPERIMENTAL SETTINGS

In this section, we introduce the main settings used in the experiments of this paper. Except for certain experiments with special hyperparameters, the same hyperparameters are adopted to ensure consistency and fairness in experimental comparisons.

### B.1  MAIN SETTING

We primarily employ the PyTorch[1], vllm[2], and Transformers[3] libraries to implement our approach, while utilizing DeepSpeed[4] for multi-GPU parallel training. All experiments are conducted on a single machine equipped with eight A800 80GB SXM GPUs. The ALPO sampling phase for each translation direction takes approximately 1.5 hours, while the training phase requires about 2 hours. To ensure fairness and consistency in comparisons, we maintain identical irrelevant parameters across the main experiments, ablation studies, and extended experiments wherever possible. All critical hyperparameter settings are presented in Table 9.

Table 9: Hyperparameter configuration in experiments.

| Phase | Hyperparameter | Value | Remark |
|---|---|---|---|
| **SFT** | n | 35 | Number of lines in a prompt $x$ |
| | optimizer | AdamW | - |
| | learning rate | 1e-6 | - |
| | epoch | 4 | - |
| | batch size | 96 | - |
| **ALPO** | $k$ | 15 | Sampling size |
| | temperature | 1.0 | |
| | top k | 40 | Sampling generation hyperparameters |
| | top p | 0.9 | |
| | optimizer | AdamW | - |
| | learning rate | 1e-6 | - |
| | epoch | 1 | - |
| | batch size | 96 | Segment-level batch size |

We re-implemented the VideoDubber method and utilized paid APIs from Alibaba Cloud, DeepSeek, OpenAI, and Anthropic to obtain experimental and evaluation results for models such as Qwen, DeepSeek, GPT, and Claude. When eliciting translations from these models on the test set via ICL, we employed prompt in Appendix F.1, with a one-shot example to ensure structured output formatting. All LLMs used in our experiments were the latest versions as of Sep 1, 2025, specifically: qwen-max-2025-01-25[5], DeepSeek-V3.1[6], gpt-4o-2024-08-06[7], GPT-5[8], and Claude 4/4.1[9].

For translation quality evaluation, we reserve 10% of each dataset's programs as the test set. Due to constraints including paid API costs and inference time, we perform sampling-based evaluation rather than full-scale assessment. From these test programs, we randomly select 2,000 subtitle segments, each containing 35 lines of subtitles, thereby constructing a test set of approximately 70,000 sentences per direction. The final evaluation results represent the average scores across these 2,000 segments.

---

[1] https://github.com/pytorch/pytorch
[2] https://github.com/vllm-project/vllm
[3] https://github.com/huggingface/transformers
[4] https://github.com/microsoft/DeepSpeed
[5] https://help.aliyun.com/zh/model-studio/what-is-qwen-llm
[6] https://api-docs.deepseek.com/news/news250821
[7] https://platform.openai.com/docs/models
[8] https://openai.com/gpt-5
[9] https://docs.anthropic.com/en/api/models-list

## B.2 Experiments in Empirical Investigation

### B.2.1 Investigation on LLM-as-a-Judge

In en⇒zh, en⇒de, zh⇒en and zh⇒th directions, we investigated the evaluation consistency of different LLMs $\pi_e$ (including DeepSeek-R1, GPT-5 Thinking, Claude Opus 4.1, and Qwen3-14B) compared to human evaluators on multiple translations of lines. Specifically, for each direction, each evaluator scored 500 lines $\{s_i \mid i \in [n], 1 \leq i \leq 500\}$ across 10 different translations on a scale of 0-100. These translations included human references, Google Translate, an SFT model $\pi_{\mathrm{sft}}$, and multiple LLMs (including DeepSeek-R1, GPT-5 Thinking, Qwen3-235B-A22B Thinking, GPT-4o, DeepSeek-V3, Claude Sonnet 4, and Qwen-Max). The SFT model $\pi_{\mathrm{sft}}$ was trained using the MuSC dataset. The evaluation LLMs assessed translations based on ICL, and since lines are typically closely related to their contextual environment, we also provided the context lines of $s_i$ to enable the LLM to reference the semantic environment of $s_i$. The prompt is detailed in Appendix F.2. The scoring criteria are:

- Principle 1 (Accuracy): The translation must ensure basic accuracy (Weight: 30%);
- Principle 2 (Colloquial Appropriateness): The translation must adhere to the expression habits of human colloquial speech (Weight: 30%);
- Principle 3 (Expressive Power): The translation needs to be expressive and vivid (Weight: 40%).

The term "weight" here serves to provide the model with an indicative reference for importance, rather than being utilized for explicit weighting operations.

For the en⇒zh and zh⇒en directions, the human evaluator group consisted of four evaluators (the same four individuals), while the en⇒de and zh⇒th directions each included two evaluators. All evaluators were undergraduate or master's students specializing in English, German, or Thai translation, with Chinese as their native language. The human scores were averaged across evaluators. In Figure 5, we verify the consistency among the four evaluators for en⇒zh and zh⇒en, confirming the reliability of the human evaluation results. For each evaluation LLM $\pi_e$, we shuffled the order of translations four times and took the average across four scores. Each evaluator assigns a score sequence $\mathcal{E}$ to multiple translations of $s_i$. Since the scores from different evaluators often fall into different ranges within the 0–100 scale, we focus only on the relative ranking of $\mathcal{E}$ rather than the absolute values. Therefore, we evaluate the Spearman rank correlation $\rho$ between different evaluators instead of using metrics such as ICC that are sensitive to absolute scores.

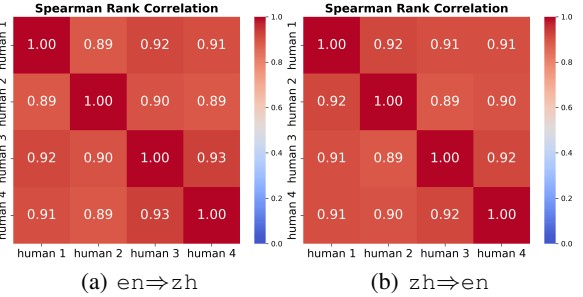

(a) en⇒zh      (b) zh⇒en

Figure 5: Human assessment consistency verification.

### B.2.2 Investigation on Back-translation Consistency

We investigated the back-translation consistency of parallel corpora from different domains. Specifically, for a translation direction, we utilized GPT-4o with a simple prompt in the following text box to translate the target language text back into the source language. We then calculated the BLEU and ChrF++ scores between the back-translated text and the original text. Lower scores indicate a higher degree of liberal translation. We evaluated 1000 sentence pairs per domain. The datasets from different domains involved in the evaluation include:

- **OpenSubtitles** (Lison & Tiedemann, 2016) is a collection of translated movie subtitles across 60 languages.
- **Books** (Tiedemann, 2012) is a collection of copyright free books in the literature domain.
- **bible-uedin** (Christodouloupoulos & Steedman, 2015) is a multilingual parallel corpus created from translations of the Bible.

- **DGT** (Steinberger et al., 2013) is a publicly accessible multilingual translation memory of the Acquis Communautaire.

- **JRC-Acquis** (Steinberger et al., 2006) is a collection of legislative texts of the European Union, comprising selected documents written from the 1950s to the present.

- **News-Commentary** (Tiedemann, 2012) is a parallel corpus in the news domain provided by WMT.

- **ECDC** (Steinberger et al., 2014) is a translation memory from the European Centre for Disease Prevention and Control.

- **EMEA** (Tiedemann, 2012) is a parallel corpus made from documents of the European Medicines Agency.

---

The en⇒de back-translation prompt demonstration for GPT-4o.

```
Please literally translate the following English text into German,
maintaining the highest degree of literal correspondence without
any localized rewriting:
Now the earth was formless and empty.  Darkness was on the surface
of the deep.  God's Spirit was hovering over the surface of the
waters.
```

---

### B.3 HUMAN TRANSLATION QUALITY EVALUATION

We conducted human evaluation of ALPO translation quality in the en⇒zh and zh⇒th directions. For each direction, we hired four evaluators, all of whom are undergraduate or master's degree professionals specializing in English or Thai translation, and whose native language is Chinese. We assessed the ALPO model against four baselines (gold reference, SFT model $\pi_{sft}$, GPT-4o, and DeepSeek-R1) across dimensions of accuracy, naturalness, and vividness, as well as through a comprehensive evaluation. The comprehensive evaluation instructions provided to the evaluators are detailed in Appendix F.3, with similar instructions for other dimensions. There were a total of 32 evaluation tasks in the multidimensional assessment across both directions. Each evaluator was assigned 4 evaluation tasks. In a single evaluation task, we provided evaluators with challenger and competitor translations of 400 subtitle segments sourced from the MuSC test set, with each segment consisting of 20 subtitle lines. Evaluators were required to choose the better translation between the two options, or mark them as "no significant difference". The subtitle segments selected for different evaluation tasks were random subsets from the test set.

## C EXTENDED EXPERIMENTS

In this section, we will present additional evaluation experiments for ALPO.

### C.1 FULL EVALUATION OF TRANSLATION QUALITY

We present the full results of the translation quality evaluation in Table 10. Our assessment of accuracy includes results from three LLM evaluators as well as traditional translation evaluation model XCOMET (Guerreiro et al., 2024). The results in Table 10 show that Google Translate achieved substantial performance in translation directions within the same language family, such as en⇒de and en⇒fr. However, its performance was poor for other languages, which might be due to the relatively simpler nature of translations between languages of the same language family. Traditional sequence-to-sequence architectures like MADLAD performed significantly worse than modern LLMs, and even VideoDubber, specifically designed for subtitle translation, did not demonstrate noticeably better performance than NLLB and MADLAD models. The comparison between ALPO and cutting-edge LLMs led to several clear conclusions:

- Different LLM evaluators provide consistent ranking of translations by varying models across multiple dimensions, further validating the conclusions in Section 3.1.

- Despite the receiving lower accuracy and naturalness scores, human translations excelled in the vividness dimension, which could indicate that human translators incorporate more interpretive translations based on the video content.
- Reason models scored higher than chat models in vividness but lower in other dimensions, suggesting that reason models adhere more to liberal translation instructions compared to chat models. This aligns with the conclusion in Section 3.3, as performing liberal translations can compromise accuracy to some extent.

## C.2 TRANSLATION DEMONSTRATION

In Table 11, we present the translation demonstrations for the en⇒zh and zh⇒en directions including human translations, chat model GPT-4o, reason model GPT-5 Thinking, and our ALPO model. It can be intuitively observed that the translations by GPT-4o tend more towards literal translation and fail to convey the emotional ambiance and stylistic information of the original subtitle through liberal translation like the other translations do. The model trained with ALPO aligns better with the subtitle translation scenario, which includes:

- The translations are more conversational, resembling human character dialogues rather than formal translations.
- The wording and phrasing are more vivid, offering expressive articulation.
- It does not excessively pursue accuracy, allowing for free translation based on context.

Furthermore, Table 11 illustrates some limitations of LLM translations. LLM translation relies solely on subtitle text and cannot reference information from video context and scenes as human translators do. For instance, LLM translates "nature" to "自然", whereas based on the video context, it should be translated to "天性". This issue surpasses what preference optimization techniques can address. Therefore, we believe multimodal machine translation might be a valuable direction for advancing subtitle translation research.

## C.3 EXTENDED ABLATION STUDY

### C.3.1 ALIGNMENT LOSS

When simplified, ALPO can be viewed as a framework compatible with various plug-and-play preference optimization losses. Specifically, the ALPO loss can be expressed as:

$$\mathcal{L}_{\text{alpo}}(\pi_\theta; \pi_{\text{ref}}) = -\mathbb{E}_{(x, \mathcal{S}(x)) \sim \mathbb{D}_{\text{alpo}}} \left( \sum_{i=1}^{n} w(s_i) \cdot \mathcal{L}_{\text{po}}(s_i) \right), \tag{7}$$

where $\mathcal{L}_{\text{po}}$ can adopt multiple types of preference optimization losses, such as DPO (Rafailov et al., 2024), SimPO (Meng et al., 2024), and GRPO (Shao et al., 2024). For the advantage-based GRPO loss, we compute it as:

$$\mathcal{L}_{\text{po}}(s_i) =$$
$$\frac{1}{|\mathcal{T}_i|} \sum_j \left[ \min \left( \frac{\pi_\theta(t_i^j \mid p_i)}{\pi_{\text{ref}}(t_i^j \mid p_i)} A_j, \text{clip}\left( \frac{\pi_\theta(t_i^j \mid p_i)}{\pi_{\text{ref}}(t_i^j \mid p_i)}, 1 - \varepsilon, 1 + \varepsilon \right) A_j \right) - \eta \text{KL}(\pi_\theta || \pi_{\text{ref}}) \right], \tag{8}$$

$$A_j = \frac{\mathcal{E}_i^j - \text{mean}(\mathcal{E}_i)}{\text{std}(\mathcal{E}_i)}, \quad \text{KL}(\pi_\theta || \pi_{\text{ref}}) = \frac{\pi_{\text{ref}}(t_i^j \mid p_i)}{\pi_\theta(t_i^j \mid p_i)} - \log \frac{\pi_{\text{ref}}(t_i^j \mid p_i)}{\pi_\theta(t_i^j \mid p_i)} - 1, \tag{9}$$

where $\varepsilon$ and $\eta$ are adjustable hyperparameters, and $A_j$ denotes the advantage estimated from the score sequence $\mathcal{E}_i$.

We compare the performance of ALPO with different preference alignment losses $\mathcal{L}_{\text{po}}$ in Table 12. The experimental results demonstrate that all loss variants achieve substantial improvements over the SFT model, validating ALPO's broad compatibility. The DPO loss exhibits stable and superior performance across multiple translation directions. While GRPO effectively utilizes all sampled outputs rather than focusing solely on preference pairs, it only achieves optimal performance in the en⇒zh direction.

Table 10: Full quality evaluation results. The 1st and 2nd best results are marked as **blue** and **orange**, respectively.

**en⇒de**

| Models | Training | Accuracy | | | | | Naturalness | | | | Vividness | | | |
|---|---|---|---|---|---|---|---|---|---|---|---|---|---|---|
| | | XCOMET | DeepSeek | Claude | GPT-5 | Avg. | DeepSeek | Claude | GPT-5 | Avg. | DeepSeek | Claude | GPT-5 | Avg. |
| Gold Reference | Human | 81.4 | 83.4 | 88.6 | 85.6 | 84.8 | 83.6 | 85.5 | 82.2 | 83.8 | 64.4 | 77.1 | 77.7 | 73.1 |
| VideoDubber | | 43.2 | 46.1 | 55.1 | 21.4 | 41.5 | 44.7 | 40.2 | 21.6 | 35.5 | 42.2 | 47.1 | 33.7 | 41.0 |
| NLLB-3.3B | ST | 70.3 | 73.5 | 88.1 | 71.4 | 75.8 | 72.2 | 75.2 | 63.0 | 70.1 | 58.2 | 60.4 | 58.9 | 59.2 |
| MADLAD-10B | | 68.9 | 70.0 | 87.3 | 66.4 | 73.2 | 68.4 | 71.1 | 57.2 | 65.6 | 55.2 | 55.7 | 52.6 | 54.5 |
| Google Translate | | 88.3 | 89.9 | 89.9 | 91.5 | 89.9 | 80.4 | 81.7 | 79.8 | 80.6 | 60.9 | 64.2 | 62.8 | 62.6 |
| GPT-4o | | 91.2 | 96.8 | 96.5 | 91.8 | 94.1 | 89.1 | 88.7 | 82.2 | 86.7 | 56.2 | 71.3 | 73.3 | 66.9 |
| Qwen-Max | ICL (C) | 91.7 | 97.8 | 97.9 | 94.7 | 95.5 | 89.6 | 90.1 | 87.8 | 89.2 | 58.0 | 74.0 | 74.7 | 68.9 |
| DeepSeek-V3.1 | | 90.2 | 97.3 | 96.9 | 94.9 | 94.8 | 89.8 | 89.9 | 87.9 | 89.2 | 56.5 | 71.4 | 74.2 | 67.4 |
| DeepSeek-R1 | ICL (R) | 89.3 | 94.4 | 94.4 | 93.0 | 92.8 | 88.0 | 88.7 | 85.5 | 87.4 | 61.1 | 74.9 | 74.1 | 70.0 |
| GPT-5 | | 90.7 | 95.7 | 94.6 | 93.4 | 93.6 | 89.1 | 89.3 | 87.5 | 88.6 | 64.8 | 76.5 | 76.9 | 72.7 |
| Qwen2.5-14B | SFT | 84.7 | 87.6 | 91.1 | 86.7 | 87.5 | 83.9 | 86.3 | 80.6 | 83.6 | 56.0 | 68.9 | 68.2 | 64.4 |
| Qwen2.5-14B | **ALPO** | 91.9 | 98.0 | 97.0 | 94.8 | 95.4 | 88.9 | 89.3 | 87.1 | 88.4 | 68.3 | 78.0 | 78.1 | 74.8 |

**en⇒fr**

| Models | Training | Accuracy | | | | | Naturalness | | | | Vividness | | | |
|---|---|---|---|---|---|---|---|---|---|---|---|---|---|---|
| | | XCOMET | DeepSeek | Claude | GPT-5 | Avg. | DeepSeek | Claude | GPT-5 | Avg. | DeepSeek | Claude | GPT-5 | Avg. |
| Gold Reference | Human | 76.4 | 84.4 | 89.0 | 84.1 | 83.5 | 85.5 | 86.8 | 82.7 | 85.0 | 68.1 | 78.6 | 77.6 | 74.8 |
| VideoDubber | | 42.3 | 46.9 | 62.1 | 41.7 | 48.2 | 48.1 | 52.5 | 41.4 | 47.3 | 47.4 | 53.4 | 44.8 | 48.5 |
| NLLB-3.3B | ST | 62.2 | 76.9 | 89.6 | 78.7 | 76.9 | 75.0 | 77.9 | 70.8 | 74.6 | 61.4 | 63.3 | 60.7 | 61.8 |
| MADLAD-10B | | 58.1 | 75.6 | 87.5 | 73.3 | 73.6 | 70.3 | 71.4 | 61.7 | 67.8 | 55.7 | 59.4 | 57.1 | 57.4 |
| Google Translate | | 81.4 | 93.5 | 95.9 | 96.7 | 91.9 | 84.5 | 85.8 | 84.2 | 84.8 | 64.3 | 64.6 | 64.0 | 64.3 |
| GPT-4o | | 86.4 | 97.7 | 96.3 | 92.6 | 93.2 | 89.8 | 90.3 | 86.4 | 88.8 | 59.0 | 74.9 | 74.7 | 69.5 |
| Qwen-Max | ICL (C) | 86.1 | 97.5 | 97.3 | 95.6 | 94.1 | 90.4 | 90.4 | 89.0 | 89.9 | 61.1 | 76.8 | 76.8 | 71.6 |
| DeepSeek-V3.1 | | 87.6 | 97.2 | 98.9 | 95.9 | 94.9 | 90.8 | 90.6 | 90.7 | 90.7 | 61.7 | 77.3 | 77.9 | 72.3 |
| DeepSeek-R1 | ICL (R) | 87.1 | 96.7 | 95.3 | 95.1 | 93.6 | 89.7 | 90.8 | 89.6 | 90.0 | 64.5 | 77.3 | 78.3 | 73.4 |
| GPT-5 | | 83.8 | 96.4 | 95.0 | 93.8 | 92.2 | 90.1 | 90.8 | 90.4 | 90.4 | 68.4 | 79.3 | 79.8 | 75.8 |
| Qwen2.5-14B | SFT | 77.7 | 89.7 | 90.6 | 86.7 | 86.2 | 85.9 | 86.8 | 83.6 | 85.4 | 57.4 | 71.7 | 74.0 | 67.7 |
| Qwen2.5-14B | **ALPO** | 86.5 | 97.9 | 96.7 | 95.4 | 94.1 | 88.7 | 91.5 | 87.4 | 89.2 | 73.7 | 82.0 | 80.7 | 78.8 |

**en⇒zh**

| Models | Training | Accuracy | | | | | Naturalness | | | | Vividness | | | |
|---|---|---|---|---|---|---|---|---|---|---|---|---|---|---|
| | | XCOMET | DeepSeek | Claude | GPT-5 | Avg. | DeepSeek | Claude | GPT-5 | Avg. | DeepSeek | Claude | GPT-5 | Avg. |
| Gold Reference | Human | 79.0 | 84.3 | 88.5 | 82.6 | 83.6 | 81.6 | 85.7 | 80.6 | 82.6 | 68.9 | 69.7 | 76.0 | 71.5 |
| VideoDubber | | 40.9 | 47.5 | 64.1 | 35.0 | 46.9 | 52.4 | 61.3 | 42.0 | 51.9 | 49.7 | 52.2 | 47.1 | 49.7 |
| NLLB-3.3B | ST | 61.9 | 65.2 | 70.4 | 48.2 | 61.4 | 59.5 | 56.8 | 45.6 | 54.0 | 46.3 | 45.2 | 39.5 | 43.7 |
| MADLAD-10B | | 53.5 | 61.0 | 79.5 | 44.7 | 59.7 | 58.3 | 65.5 | 42.7 | 55.5 | 47.4 | 46.7 | 44.8 | 46.3 |
| Google Translate | | 79.3 | 81.4 | 96.1 | 80.0 | 84.2 | 79.4 | 85.1 | 74.6 | 79.7 | 52.3 | 51.8 | 59.0 | 54.4 |
| GPT-4o | | 84.1 | 92.1 | 92.6 | 88.6 | 89.3 | 83.0 | 84.2 | 79.6 | 82.3 | 56.3 | 57.7 | 65.3 | 59.8 |
| Qwen-Max | ICL (C) | 88.3 | 95.3 | 93.5 | 90.3 | 91.9 | 85.8 | 85.8 | 81.6 | 84.4 | 59.1 | 61.7 | 63.1 | 61.3 |
| DeepSeek-V3.1 | | 85.9 | 95.6 | 93.6 | 89.5 | 91.2 | 86.9 | 86.2 | 82.8 | 85.3 | 59.5 | 61.8 | 69.2 | 63.5 |
| DeepSeek-R1 | ICL (R) | 86.9 | 92.3 | 92.9 | 90.0 | 90.5 | 87.2 | 86.0 | 83.9 | 85.7 | 68.1 | 70.0 | 74.3 | 70.8 |
| GPT-5 | | 88.4 | 96.7 | 93.6 | 90.8 | 92.4 | 88.4 | 87.1 | 85.4 | 87.0 | 69.2 | 69.9 | 74.3 | 71.1 |
| Qwen2.5-14B | SFT | 82.4 | 89.9 | 90.1 | 83.4 | 86.5 | 83.3 | 83.3 | 79.5 | 82.0 | 57.0 | 56.0 | 64.2 | 59.1 |
| Qwen2.5-14B | **ALPO** | 84.8 | 93.9 | 93.1 | 90.8 | 90.6 | 84.2 | 85.0 | 83.8 | 84.3 | 73.6 | 76.9 | 79.2 | 76.6 |

**ko⇒zh**

| Models | Training | Accuracy | | | | | Naturalness | | | | Vividness | | | |
|---|---|---|---|---|---|---|---|---|---|---|---|---|---|---|
| | | XCOMET | DeepSeek | Claude | GPT-5 | Avg. | DeepSeek | Claude | GPT-5 | Avg. | DeepSeek | Claude | GPT-5 | Avg. |
| Gold Reference | Human | 68.6 | 80.4 | 88.4 | 74.6 | 78.0 | 78.8 | 79.9 | 74.7 | 78.0 | 65.6 | 58.4 | 73.5 | 65.8 |
| VideoDubber | | 35.4 | 47.0 | 56.8 | 19.3 | 39.6 | 48.9 | 58.2 | 28.5 | 45.2 | 47.4 | 56.0 | 41.1 | 48.2 |
| NLLB-3.3B | ST | 30.9 | 39.8 | 40.1 | 21.8 | 33.1 | 30.3 | 29.5 | 18.5 | 26.1 | 27.6 | 30.6 | 17.9 | 25.4 |
| MADLAD-10B | | 37.2 | 47.6 | 66.1 | 28.8 | 44.9 | 44.7 | 57.3 | 26.8 | 42.9 | 44.4 | 55.6 | 40.2 | 46.7 |
| Google Translate | | 42.4 | 50.6 | 76.9 | 49.6 | 54.9 | 49.5 | 61.8 | 47.2 | 52.8 | 46.3 | 57.3 | 52.4 | 52.0 |
| GPT-4o | | 70.5 | 87.5 | 86.5 | 75.4 | 80.0 | 81.7 | 81.7 | 76.4 | 79.9 | 55.5 | 51.8 | 67.0 | 58.1 |
| Qwen-Max | ICL (C) | 74.5 | 89.7 | 88.8 | 81.7 | 83.7 | 83.4 | 83.7 | 80.5 | 82.5 | 57.1 | 57.6 | 70.7 | 61.8 |
| DeepSeek-V3.1 | | 73.5 | 91.9 | 87.7 | 79.1 | 83.1 | 84.7 | 82.2 | 77.8 | 81.6 | 55.6 | 49.6 | 66.4 | 57.2 |
| DeepSeek-R1 | ICL (R) | 71.1 | 86.0 | 86.4 | 75.9 | 79.8 | 84.0 | 82.3 | 78.6 | 81.6 | 64.0 | 60.0 | 72.7 | 65.6 |
| GPT-5 | | 76.3 | 93.5 | 88.7 | 79.5 | 84.5 | 85.1 | 83.2 | 79.6 | 82.6 | 63.1 | 59.3 | 72.5 | 65.0 |
| Qwen2.5-14B | SFT | 72.9 | 84.8 | 88.1 | 77.7 | 80.9 | 78.5 | 78.7 | 71.2 | 76.1 | 53.8 | 47.0 | 61.0 | 53.9 |
| Qwen2.5-14B | **ALPO** | 76.0 | 89.8 | 89.0 | 82.3 | 84.3 | 83.6 | 84.7 | 81.7 | 83.3 | 67.6 | 67.5 | 76.3 | 70.5 |

**zh⇒en**

| Models | Training | Accuracy | | | | | Naturalness | | | | Vividness | | | |
|---|---|---|---|---|---|---|---|---|---|---|---|---|---|---|
| | | XCOMET | DeepSeek | Claude | GPT-5 | Avg. | DeepSeek | Claude | GPT-5 | Avg. | DeepSeek | Claude | GPT-5 | Avg. |
| Gold Reference | Human | 78.0 | 87.8 | 86.6 | 79.4 | 83.0 | 82.0 | 81.9 | 77.1 | 80.3 | 71.9 | 71.1 | 76.9 | 73.3 |
| VideoDubber | | 54.1 | 54.3 | 66.5 | 39.5 | 53.6 | 59.2 | 60.5 | 44.6 | 54.8 | 53.9 | 52.8 | 43.6 | 50.1 |
| NLLB-3.3B | ST | 35.9 | 30.8 | 36.8 | 12.7 | 29.1 | 26.4 | 26.1 | 12.5 | 21.7 | 26.4 | 24.4 | 11.5 | 20.8 |
| MADLAD-10B | | 41.6 | 49.6 | 56.0 | 33.3 | 45.1 | 45.7 | 40.9 | 30.2 | 38.9 | 43.5 | 38.2 | 31.2 | 37.6 |
| Google Translate | | 77.2 | 75.6 | 88.1 | 78.2 | 79.8 | 70.0 | 67.6 | 61.3 | 66.3 | 56.2 | 45.4 | 49.1 | 50.2 |
| GPT-4o | | 84.8 | 92.3 | 90.3 | 86.7 | 88.5 | 85.2 | 83.9 | 79.9 | 83.5 | 61.8 | 63.4 | 68.6 | 64.6 |
| Qwen-Max | ICL (C) | 86.4 | 93.2 | 92.6 | 87.9 | 90.0 | 87.0 | 85.2 | 82.7 | 85.0 | 63.7 | 67.2 | 69.5 | 66.8 |
| DeepSeek-V3.1 | | 85.3 | 92.9 | 91.4 | 88.3 | 89.5 | 85.8 | 84.9 | 81.5 | 84.1 | 59.0 | 63.8 | 66.2 | 63.0 |
| DeepSeek-R1 | ICL (R) | 83.2 | 91.6 | 89.6 | 89.8 | 88.5 | 85.6 | 83.9 | 84.0 | 84.6 | 65.6 | 71.3 | 79.7 | 73.5 |
| GPT-5 | | 84.9 | 92.0 | 90.9 | 88.5 | 89.1 | 87.2 | 86.8 | 84.3 | 86.1 | 72.7 | 75.3 | 77.5 | 75.2 |
| Qwen2.5-14B | SFT | 80.4 | 88.7 | 88.2 | 83.5 | 85.2 | 81.9 | 82.0 | 76.3 | 80.1 | 51.5 | 52.3 | 60.7 | 54.8 |
| Qwen2.5-14B | **ALPO** | 85.1 | 90.9 | 89.8 | 87.6 | 88.3 | 86.1 | 87.1 | 87.2 | 86.8 | 79.2 | 83.0 | 82.8 | 81.7 |

**zh⇒th**

| Models | Training | Accuracy | | | | | Naturalness | | | | Vividness | | | |
|---|---|---|---|---|---|---|---|---|---|---|---|---|---|---|
| | | XCOMET | DeepSeek | Claude | GPT-5 | Avg. | DeepSeek | Claude | GPT-5 | Avg. | DeepSeek | Claude | GPT-5 | Avg. |
| Gold Reference | Human | 70.5 | 77.9 | 78.0 | 80.0 | 76.6 | 74.3 | 74.7 | 76.3 | 75.1 | 60.8 | 67.5 | 70.6 | 66.3 |
| VideoDubber | | 40.2 | 38.2 | 42.5 | 15.7 | 34.1 | 39.0 | 48.7 | 17.0 | 34.9 | 43.7 | 49.8 | 31.0 | 41.5 |
| NLLB-3.3B | ST | 41.8 | 50.2 | 59.8 | 18.7 | 42.6 | 48.9 | 41.1 | 11.7 | 33.9 | 49.3 | 41.7 | 30.5 | 40.5 |
| MADLAD-10B | | 45.6 | 51.4 | 69.0 | 25.7 | 47.9 | 54.7 | 64.7 | 33.0 | 50.8 | 52.7 | 58.1 | 42.2 | 51.0 |
| Google Translate | | 52.5 | 50.1 | 71.4 | 46.7 | 55.2 | 53.6 | 65.9 | 49.2 | 56.2 | 53.5 | 57.6 | 52.5 | 54.5 |
| GPT-4o | | 80.9 | 90.8 | 89.6 | 90.7 | 88.0 | 85.1 | 84.7 | 83.5 | 84.4 | 62.6 | 67.1 | 73.9 | 67.9 |
| Qwen-Max | ICL (C) | 84.3 | 95.1 | 91.4 | 94.5 | 91.3 | 85.9 | 86.1 | 85.5 | 85.8 | 63.8 | 70.0 | 73.6 | 69.1 |
| DeepSeek-V3.1 | | 84.8 | 92.6 | 89.3 | 93.1 | 89.9 | 85.0 | 85.2 | 83.5 | 84.6 | 61.8 | 66.6 | 72.8 | 67.1 |
| DeepSeek-R1 | ICL (R) | 82.2 | 89.2 | 88.1 | 90.8 | 87.6 | 85.1 | 84.4 | 84.0 | 84.6 | 64.4 | 72.3 | 76.2 | 71.0 |
| GPT-5 | | 83.3 | 92.9 | 88.1 | 90.5 | 88.7 | 85.1 | 82.4 | 84.1 | 83.9 | 67.3 | 73.7 | 78.1 | 73.0 |
| Qwen2.5-14B | SFT | 81.7 | 90.0 | 88.8 | 88.7 | 87.3 | 82.6 | 83.8 | 81.4 | 82.6 | 59.5 | 66.3 | 72.2 | 66.0 |
| Qwen2.5-14B | **ALPO** | 85.0 | 95.0 | 92.7 | 95.1 | 91.9 | 84.3 | 85.3 | 84.4 | 84.7 | 69.0 | 75.4 | 78.1 | 74.2 |

Table 11: Multiple translation demonstration of en⇒zh and zh⇒en. High-quality translations and inadequate translations are marked in **blue** and **orange**, respectively.

| Source | Translation | | | |
|---|---|---|---|---|
| | Human | GPT-4o | GPT-5 | ALPO (Ours) |
| Piltover was founded to escape the warmongering of mages. | 当初建立皮尔特沃夫就是为了避开那群群好战的魔法师 | 建立皮尔特沃夫正是为了逃离法师的战争 | 皮尔特沃夫是为了逃离法师的战争狂澜而建立的 | 皮尔特沃夫的创立是为了逃离术士们的穷兵黩武 |
| The Arcane is the curse of our world. My race was nearly destroyed by it. | 奥术对我们的世界来说就是诅咒 我的族人几乎因此而灭绝 | 奥术是我们世界的诅咒 我的族群几乎被它摧毁 | 奥术是我们世界的诅咒 我的族人几乎因此灭绝 | 奥术是世间的诅咒 我的族人曾险些被它摧毁殆尽 |
| Councilors Hoskel and Salo. Don't they hate each other? | 霍斯卡尔议员和萨罗议员 他俩真不都是死对头吗 | 议员们霍斯克和萨罗 他们不是互相讨厌吗 | 霍斯克尔议员和萨罗议员 他俩不是水火不容吗 | 霍斯卡尔议员和萨罗议员 他们不是水火不容吗 |
| His entire life, he's chased an impossible dream. | 他这辈子都在追逐一个不可能实现的梦想 | 他一生都在追逐一个不可能的梦想 | 他这辈子都在追逐不切实际的幻想 | 他一生都在追逐一个遥不可及的梦 |
| It was the ship of dreams to everyone else. To me, it was a slave ship. | 它是所有人心目中的梦想之船 对我它只是一条贩奴船 | 对其他人来说 这是一艘梦想之船 对我而言 它是一艘奴隶船 | 在别人眼中 这是艘梦想之船 但对我来说它却是艘奴隶船 | 对旁人来说 这艘船承载着梦想 而于我而言 它不过是艘奴隶船 |
| I would have gone overboard but Mr.Dawson here saved me. | 我差点儿摔下去 是道森先生救了我 | 如果不是道森先生救了我 我早就摔下水了 | 要不是这位道森先生救我 我差点就摔掉下去了 | 要不是道森先生及时相救 我差点葬身大海 |
| Nature has made us intolerant to change, but fortunately, we have the capacity to change our nature. | 天性会让我们排斥异变 但好在 我们有能力改变自己的天性 | 大自然让我们对变化难以接受 但幸运的是 我们有能力改变我们的本性 | 自然让我们无法适应剧变 但幸运的是 我们拥有改变天性的能力 | 自然造物使我们抗拒改变 但我们幸而拥有改变天性的能力 |
| You made me want to see the world in more than just black and white. | 你让我超越黑白分明的视角看世界 | 你让我想要从不是非黑即白的角度看世界 | 你让我学会用更丰富的眼光看世界 | 你让我渴望用缤纷的色彩 而不只是黑白两色来观照世界 |
| Don't be scared. Think of yourself as a caterpillar about to transform into a butterfly. | 别害怕 想象自己是即将变成蝴蝶的毛毛虫 | 别害怕 想象你自己是一只化蛹为蝶的毛毛虫 | 别害怕 把自己想象成即将化身成蝶的毛毛虫 | 别害怕 把自己想象成一只即将破茧成蝶的毛毛虫 |
| As long as we protected each other, we would stay immortal... and young. | 只要我们互相保护 就会永远年轻 | 只要我们保护彼此就可以 保持不死 且年轻 | 只要互相守护 就能永葆青春...与不朽 | 只要我们彼此守护 就能长生不老 永葆青春 |
| You two think you're better than us, but really, you're just a prig and a pig. | 你们俩以为自己比我们强 但你只是猪猡 | 你们俩以为自己比我们好 但事实上 你们是一只 牛鼻子老学究 和一个小猪 | 你们俩自以为高人一等 其实一个 假清高 一个 真肥猪 | 你们俩自以为高人一等 其实啊 你俩一个 道貌岸然 一个猪狗不如 |
| One day you're on top... the next, you're a clown. | 今日风光无限... 明朝小丑相见 | 有一天你在上面... 下一天你就是个小丑 | 今天你风光无限... 明天就沦为小丑 | 你今天还高高在上 明日就沦为小丑一枚 |
| Gotham loves a comeback story. | 哥谭最喜欢的就是英雄回归 | 哥谭市喜欢东山再起的故事 | 哥谭市最爱看逆袭故事 | 哥谭人最爱看咸鱼翻身的戏码 |
| You're the de facto head of the Council. People notice where you go. | 你是议会真正的领袖 行踪备受关注 | 你是议会的实际领导 人们注意你去了哪里 | 身为议会实际上的话事人 行踪自然会受关注 | 你是议会事实上的领袖 一举一动自然引人瞩目 |
| 我那个时候 就有一种强烈的预感 那个球咋 我一定能投进 | At that moment, I had a really strong intuition. That I would be able to score the shot. | At that moment, I had a strong feeling that I'd definitely make that shot. | Back then, I just had this strong feeling--that shot, you know, I was gonna make it. | At that moment, I had a strong hunch that I'd make that shot. |
| 霜华清光 凝息止气 | The luminous frost drives away all impurities. Seal her aura. | Frost's pure light, calming breath and energy. | Frostlight gathers, breath stills. | Frost Illume, still the breath, quell the air. |
| 人生很多重要的决定 都是要在这种 稀里糊涂的时候马上做的 | There are many important decisions in life that we have to make immediately when the situation is still blurry. | Many important decisions in life have to be made in these muddled moments. | Many crucial life decisions must be made precisely in these muddled moments immediately. | Many crucial decisions in life must be made on the spot, in moments of confusion and bewilderment. |
| 现在恐怕还不是 谈情说爱的时候 | I'm afraid this isn't the time for you two to be lovey-dovey. | Now's probably not the time for romance. | It's probably not the time yet to talk about love. | I'm afraid it's not the right time for lovey-dovey stuff just yet. |
| 你我皆是 被命运捉弄之人 为何不与我一起 颠覆了这世间 | Both you and I were toyed by fate. Why don't you join me in oversetting this world? | You and I are both playthings of fate. Why not join me to overthrow this world? | Both of us are victims of fate's cruelty. Why not join me to overthrow this world? | Both of us are pawns in the whims of fate. Why not join forces with me and turn this world upside down together? |
| 只可惜 这和煦的暖阳 以后再也看不到了 | It's a pity that I won't be able to feel the warm sunshine anymore. | It's such a pity that this warm sunlight, I will never see again. | It's just a pity this warm, gentle sunlight--I'll never see it again. | What a shame, I'll never get to bask in the warm sunshine ever again... |
| 你不要看我赌运差 但是我看人的眼光 可是很准的 | I might be unlucky when it comes to gambling, but I'm pretty good at judging characters. | Don't judge me by my bad luck at gambling, but my eye for people is pretty good. | Don't judge me by my bad gambling luck--my judgment of people is razor-sharp. | Just because I'm not a lucky gambler doesn't mean I'm bad at judging people. My instincts are spot-on. |
| 青山常在 绿水长流 | The world will remain the same. | The mountains are green, and the waters flow long. | The green hills endure, the blue rivers flow. | May the mountains stand tall, and the waters flow on. |
| 她为了保护你被千夫所指 因为担心你而食不下咽 | She's facing a backlash for trying to keep you safe. And she's too worried about you to eat. | She took all the blame to protect you, worried sick, unable to eat. | She endured public scorn to protect you, lost her appetite worrying about you. | She's been vilified for protecting you, and she's lost her appetite out of worry for you. |
| 她一定有三头六臂 | I'm sure she has many arms and legs. | She must have three heads and six arms. | She must have three heads and six arms or something. | She must be a superwoman. |
| 青丘军队全部停止前行 原地扎寨 不得违令 | All the Qingqiu troops shall stop advancing and camp where you are. The order must be obeyed. | The Qingqiu army must halt and set up camp immediately, no disobedience allowed. | All Qingqiu troops halt your advance, set camp immediately. Disobedience forbidden. | Qingqiu's armies, halt your advance at once. Set up camp where you stand. No disobedience allowed. |
| 人间的桂花开了 | The osmanthus in the human realm is blooming. | The osmanthus flowers are blooming in the Mortal Realm. | The osmanthus in the Mortal Realm is blooming. | The osmanthus blooms in the Mortal Realm. |

Table 12: Impact of $\mathcal{L}_{po}$. The 1st and 2nd best results are denoted as blue and orange.

| Method | $\mathcal{L}_{po}$ | en⇒de | | | en⇒zh | | | zh⇒en | | |
|--------|--------|----------|-------------|-----------|----------|-------------|-----------|----------|-------------|-----------|
| | | Accuracy | Naturalness | Vividness | Accuracy | Naturalness | Vividness | Accuracy | Naturalness | Vividness |
| SFT | - | 87.7 | 83.4 | 64.4 | 86.5 | 82.1 | 59.2 | 85.2 | 80.1 | 54.9 |
| ALPO | DPO | **95.2** | **88.3** | **74.8** | 90.6 | **84.2** | **76.6** | **88.3** | **86.8** | **81.6** |
| | SimPO | 93.2 | 87.4 | 73.9 | **90.7** | 83.9 | 74.3 | 86.2 | 84.2 | 80.9 |
| | GRPO | **94.6** | **87.9** | **74.3** | **91.2** | **85.0** | **77.1** | **87.2** | **84.9** | **81.3** |

### C.3.2 SAMPLING SIZE

The sampling size of ALPO directly affects the diversity of sampled translations. In Figure 6, we examine the impact of sampling size $k$ on translation quality in the zh⇒en direction. Results show that as $k$ increases, translation quality improves across multiple dimensions, with the most significant gain observed in vividness, and stabilizes around $k = 12$. The computation of adaptive hyperparameters $w(s_i)$ and $\beta_i$ during ALPO training depends on the number and quality variation of sampled translations. A larger sampling size directly improves the quality of training translations and leads to more suitable $w(s_i)$ and $\beta_i$. Since ALPO loss relies on the relative quality of chosen and rejected translations, performance stabilizes once the sampling size is sufficient. Based on these results, we set $k = 15$ for all other experiments.

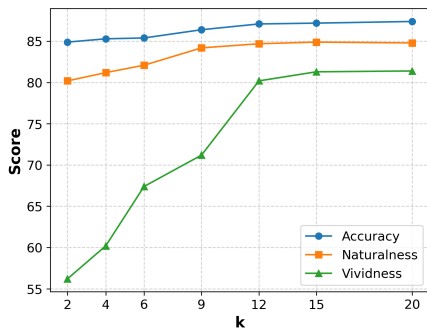

Figure 6: Impact of sampling size on translation quality.

### C.3.3 MODEL SIZE

In Figure 7, we investigate the impact of ALPO's backbone model size on translation quality. Using the Qwen2.5 series models ranging from 1.5B to 72B as ALPO's base models, we evaluate translation performance in both en⇒zh and zh⇒en directions. Experimental results demonstrate that as model size increases, scores across multiple dimensions consistently improve, though the growth rate gradually diminishes. The 14B or 32B model generally achieves performance comparable to DeepSeek-R1, further validating ALPO's significance in enhancing translation quality. ALPO employs a fully autonomous approach to preference labeling of alignment data, enabling cost-effective training of high-quality subtitle translation models. Overall, the 14B base model strikes a balanced trade-off between cost and performance. However, as subtitle translation is an offline task, larger-scale models may be adopted when pursuing SOTA performance.

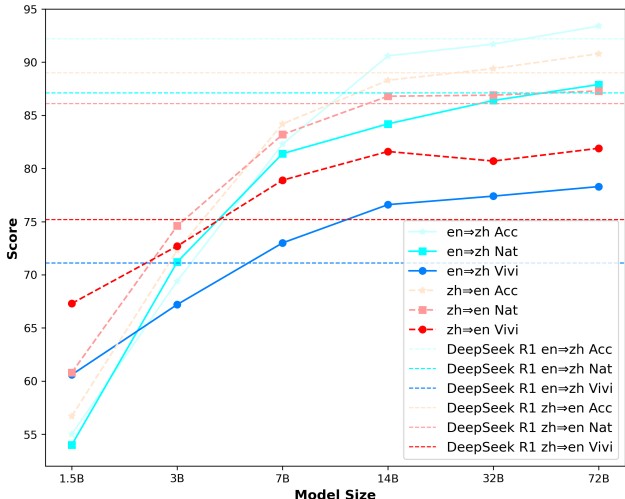

Figure 7: Impact of model size on translation quality.

### C.3.4 DATA VOLUME

We present the translation quality scores across training steps in Figure 8 to investigate the impact of training data scale. The results demonstrate that as training data volume increases, model accuracy initially rises before declining, naturalness shows modest improvement, while vividness exhibits substantial yet gradually decelerating growth. Although enhanced training data yields higher vividness scores, we observe pronounced hallucination phenomena in the model outputs, which explains the accuracy deterioration. Comprehensive analysis indicates that training for 75 steps (corresponding to 7,000 prompts with a batch size of 96) achieves optimal balance, which serves as the default configuration for all our experiments.

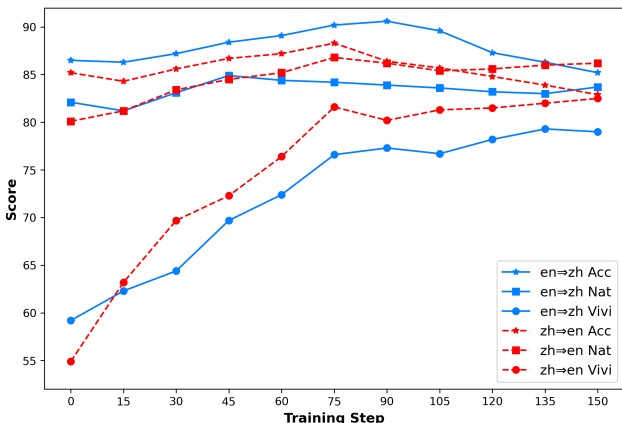

Figure 8: Impact of training data volume on translation quality.

### C.4 FURTHER EXPLORATION

### C.4.1 VANILLA DPO TRAINING

Training expressive subtitle translation LLM as a preference optimization problem, we want to explore **whether vanilla DPO can directly solve this issue.** Motivated by this inquiry, we designed and conducted relevant experiments. ALPO achieves significant improvements in translation quality through its segment-wise sampling strategy and fine-grained alignment loss, and we aim to verify the impact of these two components. Following the standard DPO training protocol (Rafailov et al., 2024), we perform post-training on the SFT model $\pi_{\text{sft}}$. Specifically, we employ either coarse-grained or fine-grained sampling to generate a chosen response $y^{(c)}$ and a rejected response $y^{(r)}$ for each sample $x \in \mathbb{D}_{\text{alpo}}$ in the alignment dataset, and subsequently optimize the policy model using the standard DPO loss.

The coarse-grained and fine-grained sampling procedures are illustrated in Algorithm 2 and Algorithm 3. In the coarse-grained sampling approach, we directly sample k complete responses for a given prompt x. Subsequently, a Qwen2-14B-Instruct model is utilized to score all subtitle lines across these k responses. The summation of scores for n subtitle lines within each response is calculated, with the highest-scoring response selected as the chosen response $y^{(c)}$ and the lowest-scoring as the rejected response $y^{(r)}$. For fine-grained sampling, we perform two separate segment-wise sampling iterations similar to Algorithm 1 for each prompt x. During these iterations, the sampling of subsequent subtitle lines employs either the highest-scored or lowest-scored existing line as the prefix, ultimately yielding the segment-wise sampled chosen response $y^{(c)}$ and rejected response $y^{(r)}$.

### C.4.2 ADVANTAGE-BASED PPO TRAINING

In the literature, it is commonly acknowledged that in preference optimization techniques, RLHF (primarily based on PPO), despite its intricate procedures, outperforms DPO (Ouyang et al., 2022; Bender et al., 2021). This motivates our investigation into **whether PPO can achieve superior performance to ALPO.** To address this, we designed and conducted relevant experiments. We adopt

---

**Algorithm 2** Coarse-grained Sampling for Vanilla DPO.

---

**Input:** SFT model $\pi_{\text{sft}}$, evaluation LLM $\pi_{\text{e}}$, alignment dataset $\mathbb{D}_{\text{alpo}}$, sample number $k$.
**Output:** sampled response pairs set $\mathcal{S}(x)$.
1: **for** any $x \in \mathbb{D}_{\text{alpo}}$ **do**    // Iterate through the alignment dataset $\mathbb{D}_{\text{alpo}}$.
2:    **for** $i = 1$ to $k$ **do**    // Sample multiple candidate responses.
3:       Sample $\pi_{\text{sft}}(y \mid x)$.
4:    **end for**
5:    Measure the sum of $\pi_{\text{e}}$ score $\mathcal{E}$ for each line of $y^i$ in the candidate set $\{y^i | i = 1, 2, \ldots, k\}$.
6:    Select chosen $y^{(\text{c})}$ and rejected $y^{(\text{r})}$.
7: **end for**
8: **return** $\mathcal{S}(x) \equiv \{y^{(\text{c})}, y^{(\text{r})}\}$.

---

**Algorithm 3** Fine-grained Sampling for Vanilla DPO.

---

**Input:** SFT model $\pi_{\text{sft}}$, evaluation LLM $\pi_{\text{e}}$, alignment dataset $\mathbb{D}_{\text{alpo}}$, sample number $k$.
**Output:** sampled response pairs set $\mathcal{S}(x)$.
1: **for** any $x \in \mathbb{D}_{\text{alpo}}$ **do**    // Iterate through the alignment dataset $\mathbb{D}_{\text{alpo}}$.
2:    **for** $i = 1$ to $n$ **do**    // The first sampling cycle used to obtain $y^{(\text{c})}$.
3:       **for** $j = 1$ to $k$ **do**
4:          Sample $\pi_{\text{sft}}(t_i^j \mid x, t_1^{(\text{c})}, \ldots, t_{i-1}^{(\text{c})})$.
5:       **end for**
6:       Deduplicate the candidate set $\{t_i^j \mid j = 1, 2, \ldots, k\}$.
7:       Add human reference $t_i^0$ to the candidate set, get $\{t_i^j \mid j = 0, 1, \ldots\}$.
8:       Measure $\{t_i^j \mid j = 0, 1, \ldots\}$ by $\pi_{\text{e}}$, get score sequence $\mathcal{E}_i$.
9:       Select a chosen translation $t_i^{(\text{c})}$ (random from **top 3** of $\mathcal{E}_i$).
10:    **end for**
11:    Concatenate $\{t_i^{(\text{c})} | i = 1, 2, \ldots, n\}$ yields $y^{(\text{c})}$.
12:    **for** $i = 1$ to $n$ **do**    // The second sampling cycle used to obtain $y^{(\text{r})}$.
13:       **for** $j = 1$ to $k$ **do**
14:          Sample $\pi_{\text{sft}}(t_i^j \mid x, t_1^{(\text{r})}, \ldots, t_{i-1}^{(\text{r})})$.
15:       **end for**
16:       Deduplicate the candidate set $\{t_i^j \mid j = 1, 2, \ldots, k\}$.
17:       Add human reference $t_i^0$ to the candidate set, get $\{t_i^j \mid j = 0, 1, \ldots\}$.
18:       Measure $\{t_i^j \mid j = 0, 1, \ldots\}$ by $\pi_{\text{e}}$, get score sequence $\mathcal{E}_i$.
19:       Select a rejected translation $t_i^{(\text{r})}$ (random from **bottom 3** of $\mathcal{E}_i$).
20:    **end for**
21:    Concatenate $\{t_i^{(\text{r})} | i = 1, 2, \ldots, n\}$ yields $y^{(\text{r})}$.
22: **end for**
23: **return** $\mathcal{S}(x) \equiv \{y^{(\text{c})}, y^{(\text{r})}\}$.

---

an advantage-based PPO training pipeline (Zheng et al., 2023b). Specifically, we implement the PPO training process through the following steps:

1. *Rollout* — Use Algorithm 4 to sample a trajectory $\tau$ for each input $x \in \mathbb{D}_{\text{alpo}}$.

2. *Computing Returns and Advantages* — Perform Generalized Advantage Estimation (GAE) on $\tau$ using the value network $V_\phi$ (base model: Qwen2.5-7B-Instruct (Yang et al., 2024a)):

$$
\begin{aligned}
\delta_i &= -\mathcal{E}_i + \gamma V_\phi(p_{i+1}) - V_\phi(p_i), \\
A_i &= \sum_{l=0}^{n-i-1} (\gamma\lambda)^l \delta_{i+l}.
\end{aligned}
\tag{10}
$$

3. *Updating Policy Network* — Let $\pi_{\text{old}}$ denote the fixed old policy during sampling. Minimize the clipped loss function (KL divergence constraint between $\pi_\theta$ and $\pi_{\text{old}}$ is omitted below):

$$\mathcal{L}_{\text{clip}}(\theta) = -\mathbb{E}_{x \sim \mathbb{D}_{\text{alpo}}} \left[ \sum_{i=1}^{n} \min \left( \frac{\pi_\theta(t_i \mid p_i)}{\pi_{\text{old}}(t_i \mid p_i)} A_i, \text{clip} \left( \frac{\pi_\theta(t_i \mid p_i)}{\pi_{\text{old}}(t_i \mid p_i)}, 1 - \epsilon, 1 + \epsilon \right) A_i \right) \right]. \tag{11}$$

4. *Updating Value Network* — Fit $V_\phi(p_i)$ to the GAE-based estimate $\hat{V}_i = A_i + V_\phi(p_i)$ via mean squared error:

$$\mathcal{L}_V(\phi) = \mathbb{E}_{x \sim \mathbb{D}_{\text{alpo}}} \left[ \sum_{i=1}^{n} (V_\phi(p_i) - \hat{V}_i)^2 \right]. \tag{12}$$

5. *Repeat for Multiple Epochs* — Continuously collect new data and update the value and policy networks until convergence.

---

**Algorithm 4** Sampling for PPO solution.

---

**Input:** SFT model $\pi_{\text{sft}}$, evaluation LLM $\pi_{\text{e}}$, alignment dataset $\mathbb{D}_{\text{alpo}}$.
**Output:** sampled trajectory set $\mathcal{S}(x) \equiv \{\tau\}$.
1:  **for** any $x \in \mathbb{D}_{\text{alpo}}$ **do**    // Iterate through the DA dataset $\mathbb{D}_{\text{alpo}}$.
2:      **for** $i = 1$ to $n$ **do**    // Iterate through the subtitle lines in $x$.
3:          Sample $\pi_{\text{sft}}(t_i \mid x, t_1, \ldots, t_{i-1})$.
4:          Measure $t_i$ by $\pi_{\text{e}}$, get score $\mathcal{E}_i$.
5:      **end for**
6:      Obtain trajectory $\tau = \{(p_i, t_i, \mathcal{E}_i) \mid i = 1, 2, \cdots, n\}$.
7:  **end for**
8:  **return** $\mathcal{S}(x) \equiv \{\tau\}$.

---

### C.4.3 EVALUATION

We evaluated the performance of en⇒zh and zh⇒en translation under "Vanilla DPO Training" and "Advantage-based PPO Training" configurations using the Qwen2.5-14B model. In addition, we also compared two token-level DPO-based methods, TDPO (Zeng et al., 2024) and TIS-DPO (Liu et al., 2025). Experimental results presented in Table 13 demonstrate that while DPO and PPO training with segment-level sampling strategy achieved moderate performance improvements compared to the SFT model, they still exhibited significant gaps compared to ALPO with fine-grained alignment loss. ALPO also significantly outperformed the two token-level methods. This validates the effectiveness of both segment-wise sampling strategy and fine-grained alignment loss for the local preference optimization task.

Table 13: Experimental evaluation results of alternative solutions. C for Coarse-grained, while F for Fine-grained. The 1st and 2nd best results are denoted as **blue** and **orange**.

| Model | Train | Sampling | en⇒zh | | | zh⇒en | | |
|---|---|---|---|---|---|---|---|---|
| | | | Accuracy | Naturalness | Vividness | Accuracy | Naturalness | Vividness |
| Gold Reference | - | - | *83.6* | *82.9* | *71.6* | *82.9* | *80.2* | *73.2* |
| GPT-4o | - | - | *89.5* | *82.4* | *59.5* | *88.6* | *82.9* | *64.6* |
| DeepSeek-R1 | - | - | *90.4* | *85.6* | *70.7* | *88.5* | *85.6* | *73.6* |
| Qwen2.5-14B | SFT | - | 86.5 | 82.1 | 59.2 | 85.2 | 80.1 | 54.9 |
| | DPO | C | 87.1 | 82.2 | 63.3 | 86.1 | 81.1 | 60.2 |
| | | F | 88.2 | 82.5 | 69.4 | **86.3** | 83.1 | 68.4 |
| | PPO | F | 88.3 | **83.2** | **70.1** | 85.9 | 82.3 | **70.1** |
| | TDPO | F | 87.4 | 82.7 | 64.0 | 86.0 | 81.3 | 61.2 |
| | TIS-DPO | F | **88.5** | 83.1 | 65.9 | 86.2 | **83.3** | 62.4 |
| | **ALPO** | F | **90.6** | **84.2** | **76.6** | **88.3** | **86.8** | **81.6** |

### C.5 FURTHER APPLICATION OF ALPO IN MULTI-TURN INTERACTION

### C.5.1 OVERVIEW

To validate the generality of the ALPO method, we conducted experiments on another local preference optimization task: social language agent multi-turn interaction (Zhou et al., 2024; Wang et al., 2024a;

Lu et al., 2024; Kong et al., 2025). This task requires LLM agents to dynamically adjust generation strategies through multi-turn interactions in an interactive environment, guided by character profiles, contextual scenarios, and private social objectives. The objective is to enable the agent to more effectively accomplish predefined goals (e.g., persuasion, agreement attainment) within specific social scenarios (e.g., negotiation, collaboration, competition), while simultaneously maintaining or improving the relationship between both dialogue parties.

### C.5.2 METHOD

For the multi-turn interaction task, we primarily complete preference alignment data sampling through two steps: 1) Error Location; 2) Preference Data Sampling. For a multi-turn dialogue in the training set, we first employ GPT-4o to identify the starting position of erroneous turns, following the same criteria as Kong et al. (Kong et al., 2025): 1) The turn is critical to achieving the role's goal; 2) There remains room for improvement in the relationship between goal completion and the role.

If both criteria are satisfied, we take the dialogue history preceding this turn as the initial sampling prefix $p_1$, then utilize our LLM agent to sequentially sample 5 single-turn responses for each turn. During the $i$-th sampling iteration, we employ GPT-4o to determine whether the current turn qualifies as crucial based on the sampled response: responses containing primarily pleasantries or similar content are deemed non-crucial, with their indicator function $\mathbf{1}(p_i)$ set to 0; otherwise, it is set to 1. We then select the response with the highest combined goal and relationship score as the preferred response $y_i^w$, and the lowest as the dispreferred response $y_i^l$, with goal completion prioritized over relationship. The prefix $p_i$ for the $i$-th iteration becomes $p_{i-1}, y_{i-1}^w$ (including interlocutor behaviors, omitted here). Finally, we apply the following simplified ALPO loss function for alignment training:

$$\mathcal{L}_{\text{alpo}}(\pi_\theta; \pi_{\text{ref}}) = -\mathbb{E}_{(x, y_{1:n}^w, y_{1:n}^l) \sim \mathbb{D}} \left[ \sum_{i=1}^{n} \mathbf{1}(p_i) \cdot \log \sigma \left( \beta \log \frac{\pi_\theta(y_i^w \mid p_i)}{\pi_{\text{ref}}(y_i^w \mid p_i)} - \beta \log \frac{\pi_\theta(y_i^l \mid p_i)}{\pi_{\text{ref}}(y_i^l \mid p_i)} \right) \right],$$
(13)

where $\mathbb{D}$ denotes the dataset, and both $\pi_\theta$ and $\pi_{\text{ref}}$ are initialized from behavioral cloning (BC) models.

### C.5.3 EXPERIMENTS

We utilize 100 out of the 410 scenarios from SOTOPIA-$\pi$ (Wang et al., 2024a) for behavioral cloning (10 role pairs per scenario) and 310 scenarios for alignment (8 role pairs per scenario). We adopt SOTOPIA (Zhou et al., 2024) as the test set, containing 90 scenarios with 5 role pairs per scenario, totaling 450 self-chat tasks and 900 non-self-chat tasks. We evaluate both goal and relationship dimensions on the same baselines, with experimental results presented in Table 14. Baseline results are sourced from Kong et al. (Kong et al., 2025).

Table 14: The performance of various methods on SOTOPIA across the goal and relationship dimensions. The 1st and 2nd best results are denoted as **blue** and **orange**, respectively.

| Methods | Self-Chat | | GPT-4o | |
|---|---|---|---|---|
| | Goal | Rel | Goal | Rel |
| GPT-3.5-turbo | 6.38 | 1.36 | 7.19 | 2.05 |
| GPT-4o-mini | 6.98 | 2.11 | 7.44 | 2.36 |
| GPT-4-turbo | 8.18 | 2.96 | 7.92 | 2.79 |
| GPT-4o | 7.90 | 2.67 | 7.90 | 2.67 |
| LLaMA-8B | 7.24 | 1.94 | 7.70 | 2.49 |
| LLaMA-8B+BC | 7.81 | 3.05 | 7.53 | 2.78 |
| LLaMA-8B+BC+Preferred-SFT | 7.76 | 3.05 | 7.65 | 2.88 |
| LLaMA-8B+BC+DPO (Rafailov et al., 2024) | 7.95 | 3.28 | 7.80 | 2.97 |
| LLaMA-8B+BC+ETO (Song et al., 2024) | 8.29 | 3.39 | 8.02 | 3.03 |
| LLaMA-8B+BC+DMPO (Shi et al., 2024) | 8.28 | 3.37 | 8.00 | 2.98 |
| LLaMA-8B+BC+SDPO (Kong et al., 2025) | **8.56** | **3.69** | **8.13** | **3.16** |
| LLaMA-8B+BC+ALPO | **8.41** | **3.56** | **8.21** | **3.19** |

The experimental results demonstrate the effectiveness of ALPO on other application tasks. ALPO employs a progressive optimization strategy transitioning from local to global optima, while utilizing

an indicator function to adaptively determine the participation of each segment in the loss computation. This approach enables effective handling of multi-segment local optimization tasks.

# D  THEORY: ADAPTIVE LOCAL PREFERENCE OPTIMIZATION

In this section, we formalize the local preference optimization problem and validate the effectiveness of ALPO and the limitations of general preference optimization approaches. Note that for the sake of expository convenience, the notations used in this section may differ from those in previous sections.

## D.1  LOCAL PREFERENCE OPTIMIZATION PROBLEM

For a given input $x \in \mathcal{X}$ ($\mathcal{X}$ denotes the input space) to the language model $\pi_\theta$, assume it consists of $n$ interrelated segments, i.e., $x = (x_1, \cdots, x_n)$. Correspondingly, its output $y \in \mathcal{Y}$ ($\mathcal{Y}$ being the output space) is also composed of $n$ interrelated segments, i.e., $y = (y_1, \cdots, y_n)$, where $x_i$ and $y_i$ exhibit a one-to-one correspondence. Additionally, the generation of $y_i$ is influenced by $y_1, \cdots, y_{i-1}$ (note that this influence arises not only from the autoregressive nature of language model but also potentially from semantic dependencies between output segments, among other factors), i.e.:

$$\pi_\theta(y \mid x) = \prod_{i=1}^{n} \pi_\theta(y_i \mid x, y_1, \cdots, y_{i-1}). \tag{14}$$

Consider a segment-level preference metric $r(x_i, y_i)$ (a reward signal or other quantitative measure) that evaluates how well the segment output $y_i$ aligns with the optimization objective for its corresponding segment input $x_i$. The goal of the local preference optimization problem is to adjust the parameters $\theta$ such that $\pi_\theta$ is optimized to maximize $r(x_i, y_i)$. This objective is formally expressed as:

$$\theta^* = \arg \max_\theta \mathbb{E}_{x \sim p(x)} \Big[ \mathbb{E}_{y \sim \pi_\theta(\cdot \mid x)} \big[ \sum_{i=1}^{n} r(x_i, y_i) \big] \Big], \tag{15}$$

where $p(x)$ represents the true distribution of the input $x$.

## D.2  INEFFECTIVENESS OF OUTCOME-SUPERVISED PREFERENCE OPTIMIZATION METHODS

We take DPO (Rafailov et al., 2024) as an example to illustrate the limitations of general preference optimization methods when addressing local preference optimization problems. When performing local preference optimization, DPO first annotates preferred responses $y^w = (y_1^w, \cdots, y_n^w)$ and non-preferred responses $y^l = (y_1^l, \cdots, y_n^l)$ corresponding to $x = (x_1, \cdots, x_n)$ based on $r(x_i, y_i)$. These responses inherently satisfy:

$$\pi_\theta(y^w \mid x) = \prod_{i=1}^{n} \pi_\theta(y_i^w \mid x, y_1^w, \cdots, y_{i-1}^w), \pi_\theta(y^l \mid x) = \prod_{i=1}^{n} \pi_\theta(y_i^l \mid x, y_1^l, \cdots, y_{i-1}^l). \tag{16}$$

Then, DPO optimizes $\pi_\theta$ through a single-stage policy optimization approach:

$$\mathcal{L}_{\text{DPO}}(\pi_\theta; \pi_{\text{ref}}) = -\mathbb{E}_{(x, y^w, y^l) \sim \mathbb{D}} \left[ \log \sigma \left( \beta \log \frac{\pi_\theta(y^w \mid x)}{\pi_{\text{ref}}(y^w \mid x)} - \beta \log \frac{\pi_\theta(y^l \mid x)}{\pi_{\text{ref}}(y^l \mid x)} \right) \right], \tag{17}$$

where $\pi_{\text{ref}}$ denotes a frozen reference model. For the contrastive terms in the loss function, compute:

$$\log \frac{\pi_\theta(y^w \mid x)}{\pi_{\text{ref}}(y^w \mid x)} - \log \frac{\pi_\theta(y^l \mid x)}{\pi_{\text{ref}}(y^l \mid x)} = \sum_{i=1}^{n} \left[ \log \frac{\pi_\theta(y_i^w \mid x, y_{1:i-1}^w)}{\pi_{\text{ref}}(y_i^w \mid x, y_{1:i-1}^w)} - \log \frac{\pi_\theta(y_i^l \mid x, y_{1:i-1}^l)}{\pi_{\text{ref}}(y_i^l \mid x, y_{1:i-1}^l)} \right], \tag{18}$$

where $y_{1:i-1}^w = (y_1^w, \cdots, y_{i-1}^w)$ and $y_{1:i-1}^l = (y_1^l, \cdots, y_{i-1}^l)$. The limitations of this approach are:

- **Lack of Fine-Grained Contrast on Aligned Prefixes**: The generation probability of the $i$-th segment is conditioned on its own preferred or dispreferred prefix ($y_{1:i-1}^w$ or $y_{1:i-1}^l$), rather than a shared prefix. If $y_i^w$ and $y_i^l$ exhibit significant divergence in their prefixes (e.g., differing styles or topic branches), the contrast at the $i$-th segment ceases to be a fair comparison under identical conditions.

- **Inability to Apply Training Signals Per Segment**: DPO performs a holistic preference judgment over complete sequences $(y^w, y^l)$, depriving the model of segment-level corrective guidance. Specifically, while the model recognizes the global superiority of $y^w$ over $y^l$, it lacks localized feedback to optimize decisions at individual segments.
- **Gradient Dilution and Noise**: When the divergence between $y^w$ and $y^l$ occurs only at specific segments, the global contrast aggregates gradients from numerous irrelevant segments (nearly identical in generation). This drowns critical alignment signals, hindering effective identification and correction of misaligned segments.

In summary, DPO underperforms in local preference optimization tasks due to misaligned generative conditions caused by prefix divergence and the absence of stepwise "correct vs. incorrect" decision learning (a limitation shared by other outcome-supervised preference optimization losses).

### D.3 ADAPTIVE LOCAL PREFERENCE OPTIMIZATION

Unlike vanilla DPO, ALPO labels the preferred response $y_i^w$ and dispreferred response $y_i^l$ for each segment using $y_{1:i-1}^w$ as the prefix, instead of generating the entire response sequence. During training, the prefix for the $i$-th segment is fixed as $p_i = (x, \hat{y}_1, \cdots, \hat{y}_{i-1})$, where $\hat{y}_i$ is obtained through the scheduled prefix mixing strategy. This ensures that comparisons for each segment are made under the same preferred prefix, eliminating unfair competition between "preferred prefix vs. dispreferred prefix". Under the same prefix, the preference alignment loss contrasts $\pi_\theta(y_i^w \mid p_i)$ and $\pi_\theta(y_i^l \mid p_i)$ conditioned on $p_i$. Finally, the summation of segment-wise losses imposes preference constraints at each segment:

$$\mathcal{L}_{\text{alpo}}(\pi_\theta; \pi_{\text{ref}}) = -\mathbb{E}_{(x, y_{1:n}^w, y_{1:n}^l) \sim \mathbb{D}} \left[ \sum_{i=1}^{n} w(p_i) \cdot \log \sigma \left( \beta_i \log \frac{\pi_\theta(y_i^w \mid p_i)}{\pi_{\text{ref}}(y_i^w \mid p_i)} - \beta_i \log \frac{\pi_\theta(y_i^l \mid p_i)}{\pi_{\text{ref}}(y_i^l \mid p_i)} \right) \right], \tag{19}$$

where $w(p_i)$ and $\beta_i$ are adaptive hyperparameters that determine the contribution of each segment to the optimization process based on specific criteria.

The reason for using scheduled prefix mixing to obtain $\hat{y}_{1:i-1}$ as the prefix is that, as optimization proceeds, the model tends to generate $y_i^w$, making the post-training distribution more consistent with the $y_{1:i-1}^w$ path. Since $y_{1:i-1}^w$ is also the prefix path most likely to be followed, this ensures that the model learns to generate the preferred segment $y_i^w$ rather than $y_i^l$ under the prefixes it visits most frequently. This aligns with a common principle in reinforcement learning and preference optimization: *the state (or prefix) distribution during training should match the distribution most likely to be visited by the final policy (on-policy)* (Christiano et al., 2017; Sutton, 2018; Ouyang et al., 2022). Formally, during training we aim to minimize:

$$\mathbb{E}_{p \sim d_{\theta^*}} \left[ \mathcal{L}_{\text{alpo}} \left( \pi_\theta(\cdot \mid p) \right) \right], \tag{20}$$

where $d_{\theta^*}$ denotes the prefix distribution induced by the final model. Since $d_{\theta^*}$ is likely to include preferred prefixes upon convergence, selecting $\hat{y}_{1:i-1}$ as the prefix essentially samples (or directly uses) states from the preferred path, ensuring consistency with the final distribution $d_{\theta^*}$.

## E DISCUSSION

In this section, we provide further discussion on the ALPO method.

### E.1 CONCLUSION

The visual media industry, serving as a pivotal medium for human cultural exchange and dissemination, is witnessing automation and industrialization as critical future trends. Against this backdrop, we propose ALPO, a novel training paradigm for translation models, by employing techniques including LLM-as-a-Judge and preference alignment. Through comprehensive experimentation and theoretical analysis, we validate the effectiveness of ALPO. We believe this study will not only positively contribute to the advancement of visual media technologies but also promote research on domain-specific translation models in other fields.

## E.2 RECOMMENDATIONS

We would like to offer some developer suggestions for technicians using ALPO:

- When employing LLMs for preference annotation, shuffle the order of translations across multiple inference passes and average the results to obtain robust evaluation outcomes.
- The training prompt should utilize the language best suited to the selected backbone, e.g., Chinese prompts for Qwen-series models and English prompts for LLaMA-series models.
- Our open-source implementation reveals additional processing details, including handling of terminology translation, subtitle segmentation, and semantic integrity preservation. Refer to the source code for specific implementations.

Subtitle texts are inherently multimodal content closely intertwined with the visual and auditory elements of media programs. Videos provide supplementary information about characters, scenes, and plot progression, while audio conveys emotional and tonal cues. Future research should therefore focus on harnessing these multimodal data resources to enhance subtitle translation, e.g., leveraging video and audio modalities to enable speaker annotation for subtitle lines, and utilizing audiovisual context to facilitate subtitle translation.

## E.3 LIMITATIONS

ALPO has the following limitations:

- The visual media data and associated copyrights required for subtitle translation are predominantly concentrated within large enterprises, creating data accessibility barriers.
- Similar to other SOTA LLMs, ALPO lacks the capability to integrate multimodal information (e.g., video) during the translation process.

## F PROMPTS AND INSTRUCTIONS

In this section, we present the input and output formats used for LLM and human evaluators.

### F.1 INPUT AND OUTPUT OF TRANSLATION LLM

We illustrate the prompt and response format of the zh⇒en translation model below (similar formats apply to other languages). The red text appears only in LLM ICL. The original and translated subtitles of the program are formatted as shown in the text boxes for use in SFT and ALPO model training, as well as other baseline LLM ICL.

---

**The zh⇒en translation prompt demonstration of translation model.**

```
<INSTRUCTION>
Please translate the following multiple Chinese movie/TV lines into
English according to the following requirements:
1.  The English translation should be colloquial, easily
understandable, and that the language style is consistent with the
Chinese lines.
2.  The English translation must be expressive and vivid,
effectively conveying the atmosphere, emotions, and tone of the
original Chinese lines.
3.  Output the translation with its line number, ensuring that the
number of lines in the translation matches the number of lines in
the Chinese original, without merging any lines.

<EXAMPLE>
Original text:
1.请给你我一点和平相处的时间
2.我们必须谈谈
...
```

---

```
34.奥术在你体内涌动
35.请你离海克斯飞门远点

Translation results:
1.Allow us a moment of civility.
2.We must talk.
...
34.The Arcane stirs within you.
35.Stay away from the Hexgates.

<TASK>
According to the requirements, complete the following translation.
Original text:
1.你的病情我都听说了
2.我真的很遗憾 孩子
...
34.令人心痛的是 越是光芒万丈的人
35.越会更快地燃尽生命的能量

Translation results:
```

The zh⇒en translation response demonstration of SFT model.

```
1.I heard your prognosis.
2.I'm terribly sorry, my boy.
...
34.It's a sad truth that
35.those who shine brightest often burn fastest.
```

## F.2 INPUT OF EVALUATION LLM

We present the prompt format for LLM evaluation of vividness in zh⇒en translation below (similar formats apply to other languages). When using LLM to assess the vividness of a line, we provide the contextual lines surrounding it.

The zh⇒en prompt demonstration of evaluation LLM.

```
<INSTRUCTION>
Please assign a vividness score (0-100, integer) to multiple
English translations of a Chinese subtitle line.  The original
Chinese line and its surrounding context will be provided for
reference, marked with [To be evaluated] and [Context] respectively.
The scoring principles for translation vividness are as follows:
Principle 1 (Accuracy):  The translation can be moderately liberal
but must maintain reasonable accuracy in conveying the original
meaning without additions or omissions (Weight:30%);
Principle 2 (Colloquial Appropriateness):  Evaluate whether the
translation uses natural spoken language suitable for the subtitle
context and effectively conveys the original emotion, atmosphere,
and tone (Weight:30%);
Principle 3 (Expressive Power):  Assess whether the translation
is emotionally resonant, employs vivid phrasing, and has literary
qualities that engage the audience (Weight:40%).

Scoring Criteria (Vividness):
100:  Exceptionally expressive and impactful translation with rich
emotional layers that deeply resonate with the audience.
50:  Moderately expressive translation with subdued emotional
delivery that partially conveys the intended feelings.
0:  Inaccurate translation lacking emotional depth or expressive
```

```
qualities, failing to evoke any audience connection.

<EXAMPLE>
Chinese original text:
[Context] 被你到处践踏的遗迹 可是无价之宝
[Context] 是无法用价值来衡量的珍宝
[To be evaluated] 历史虽然会重演 但是人类是无法回到过去的
[Context] 看来你是不会明白的
[Context] 明...明白...我明白了

English translation:
Translation A: History repeats, but we can't go back to what was.

Translation B: History often echoes, yet there's no way for us to
turn back the clock.

...

Evaluation score:
{"A":  70, "B":  92, "C":  85, ...}

<TASK>
According to the criterion, complete the following evaluation.
Chinese original text:
[Context] 我的贡献注定是转瞬即逝 连您都会很快遗忘
[Context] 我见过的学生很多
[To be evaluated] 令人心痛的是 越是光芒万丈的人 越会更快地燃尽生命的能量
[Context] 我不知道你还是个艺术家
[Context] 我的事你不知道的多着呢

English translation:
Translation A: It's a sad truth that those who shine brightest
often burn fastest.

Translation B: The heartbreaking truth is that those who shine the
brightest tend to burn through their life's energy all the faster.

...

Note, you need to output the ratings in JSON format:  {"A":  score,
"B":  score, "C":  score, ...}
Evaluation score (only output the rating, no other content):
```

### F.3 INSTRUCTION AND PROMPT FOR QUALITY EVALUATION

For human evaluation of translation quality, it is essential to provide evaluators with clear instructions specifying the evaluation perspectives, criteria, and format. These instructions directly influence the focus and emphasis of evaluators during the quality assessment process. The instructions provided to evaluators are shown in the first text box, while the prompts used for multidimensional automated evaluation with LLMs are presented in the three subsequent boxes.

**Instruction for human evaluation of translation quality.**

```
[Evaluation Criteria]

1.  Accuracy
When evaluateing the accuracy of audiovisual subtitle translation,
consider the following dimensions:
    •Semantic Equivalence:  Evaluate whether the meaning of the
original subtitle is accurately conveyed in the translated version,
and if the semantic content of the source subtitle is precisely
```

expressed in the target subtitle.
   ·**Grammatical Correctness**: Evaluate the grammatical accuracy of the translated subtitle, including sentence structure, tense, voice, and other grammatical aspects.
   ·**Terminology Translation**: Evaluate whether proper nouns have been accurately translated, maintaining the semantics and context of the original terms.
**2. Naturalness**
When evaluateing the naturalness of audiovisual subtitle translation, consider the following dimensions:
   ·**Coherence**: Evaluate whether the translated subtitle reads as if written by a native speaker of the target language, and evaluate the logical relationships between sentences.
   ·**Readability**: Evaluate whether the translated subtitle is easy to read and understand, and if the word choice and expressions conform to the conventions of the target language.
   ·**Fluency**: Evaluate whether the translated subtitle flows smoothly, if sentences are well-constructed, and if there are any obvious grammatical errors or unnatural expressions.
**3. Vividness**
When evaluateing the vividness of audiovisual subtitle translation, consider the following dimensions:
   ·**Stylistic Consistency**: Evaluate whether the translated subtitle maintains the style and characteristics of the original, including consistency in character tone and emotional nuances.
   ·**Expressiveness**: Evaluate whether the translation conveys the essence and atmosphere of the original lines, avoiding mechanical literal translation, thereby making it easier for the audience to understand and find engaging.
   ·**Emotion**: Assess whether the translation faithfully conveys the character's emotions, aligns with the scene and character context, and resonates emotionally with the target language audience.

**[Task]**

For each set of original subtitles, two different translations (A and B) are provided. Please refer to the multiple evaluation dimensions specified in the [Evaluation Criteria] to evaluate the two translations for each set of original subtitles. Indicate your evaluation results for translations A and B by marking [A is better], [B is better], or [No significant difference between A and B]. Note that you only need to evaluate the overall performance of each set of subtitles, not each individual line of subtitle.

---

The prompt of en⇒zh subtitle translation accuracy evaluation.

[English to Chinese Subtitle Translation Accuracy Evaluation]

Please rate the accuracy of the following English to Chinese subtitle translation, using integer scores from 0 to 100. The translation accuracy rating criteria include evaluating whether the Chinese translation accurately conveys the original meaning of the English subtitle. Additionally, pay attention to whether the terminology (e.g., names of people, places, organizations, items, etc.) in the English original subtitle is accurately translated in the Chinese translation.

Scoring Criteria (Accuracy):
100: The translation is completely accurate, covers all information, provides a coherent translation, and the proper nouns are accurately translated.
50: The translation is mostly accurate, with only minor omissions

```
or unclear context, and proper nouns are partially inaccurately
translated, but the overall meaning is still understandable.
0:  The translation is severely distorted, misinterprets the main
meaning of the original text, and the translation of proper nouns
is poor.

Original English Line:
A crime like this can't be overlooked.  The boy must be punished.
A violation of the Ethos calls for banishment,
but I can sympathize with a young man's dream to change the world.
...
Yeah, I must admit, his theory intrigues.
If dangerous ideas didn't excite the imagination,
we would never wander astray.

Chinese Translation:
这样严重的罪行不能轻易放过
违反社会共识的人确实应该遭到驱逐
但我也能体会一个年轻人梦想改变世界的雄心
...
我不得不说 他那套学说非常有意思
如果危险的念头不曾引人遐想
也就没有误入歧途一说了

Note:  You need to output in JSON format: {"Score":  evaluation
score}
Score (only output the score, no further explanation required):
```

---

**The prompt of en⇒zh subtitle translation naturalness evaluation.**

```
[English to Chinese Subtitle Translation Naturalness Evaluation]

Please rate the naturalness of the following English to Chinese
subtitle translation using an integer score from 0 to 100.  The
naturalness score of the subtitle translation should consider
whether the translated text adequately takes into account
contextual factors, including cultural background and context,
and whether it ensures natural and fluent language expression
that conforms to Chinese grammatical structures and word usage
habits.  It should be easy to understand and close to the culture
and expression habits of the Chinese audience.

Scoring Criteria (Naturalness):
100:  The translation fully considers context and cultural
background, with smooth and natural language that aligns with
Chinese usage habits, and contains no grammatical or lexical
errors.
50:  The translation considers the context and is basically fluent,
but some expressions may be slightly awkward or unnatural, with
potential minor grammatical errors.
0:  The translation does not effectively consider the context or
culture, is not fluent, has many grammatical errors, is rigid, and
difficult to understand.

Original English Line:
A crime like this can't be overlooked.  The boy must be punished.
A violation of the Ethos calls for banishment,
but I can sympathize with a young man's dream to change the world.
...
Yeah, I must admit, his theory intrigues.
If dangerous ideas didn't excite the imagination,
we would never wander astray.
```

```
Chinese Translation:
这样严重的罪行不能轻易放过
违反社会共识的人确实应该遭到驱逐
但我也能体会一个年轻人梦想改变世界的雄心
...
我不得不说 他那套学说非常有意思
如果危险的念头不曾引人遐想
也就没有误入歧途一说了

Note:  You need to output in JSON format:  {"Score":  evaluation
score}
Score (only output the score, no further explanation required):
```

**The prompt of en⇒zh subtitle translation vividness evaluation.**

```
[English to Chinese Subtitle Translation Vividness Evaluation]

Please rate the vividness of the English to Chinese subtitle
translation below, using an integer score from 0 to 100.  The score
for translation vividness does not take into account the accuracy
of the translation; it only evaluates whether the translation is
expressive, emotionally rich, and more capable of engaging the
audience.

Scoring Criteria (Vividness):
100:  The translation is highly expressive and impactful, with rich
emotions, capable of strongly moving the audience.
50:  The translation has some expressiveness, and the emotional
expression is relatively flat but still conveys some emotion.
0:  The translation lacks expressiveness and emotion, failing to
evoke any emotional resonance from the audience.

Original English Line:
A crime like this can't be overlooked.  The boy must be punished.
A violation of the Ethos calls for banishment,
but I can sympathize with a young man's dream to change the world.
...
Yeah, I must admit, his theory intrigues.
If dangerous ideas didn't excite the imagination,
we would never wander astray.

Chinese Translation:
这样严重的罪行不能轻易放过
违反社会共识的人确实应该遭到驱逐
但我也能体会一个年轻人梦想改变世界的雄心
...
我不得不说 他那套学说非常有意思
如果危险的念头不曾引人遐想
也就没有误入歧途一说了

Note:  You need to output in JSON format:  {"Score":  evaluation
score}
Score (only output the score, no further explanation required):
```

