# OpenReview forum: "From Utterance to Vividity: Training Expressive Subtitle Translation LLM via Adaptive Local Preference Optimization"
_ICLR.cc/2026/Conference — ICLR 2026 Poster_

### Official Review · Reviewer_i1xn · 2025-10-31

**Soundness:** 4
**Presentation:** 4
**Contribution:** 3
**Rating:** 10
**Confidence:** 5

**Summary:**

The paper describes in detail a method to adapt LLMs for subtitle translation. The paper includes a number of preliminary studies that motivate the specific challenges of this task (such as that vividity and naturalness are more important than literalness), and the use of LLM as a judge to assess vividity. It then puts all these insights into practice into a well developed multi-step process, including a new preference optimization method. This is a very well written paper.

**Strengths:**

Very well written and motivated work, well executed solution.

Insightful analysis into what makes subtitle translation different from more factual translation challenges such as in the news/legal domain.

**Weaknesses:**

The correlation assessment would benefit from a second human assessment to get a sense of inter-annotator agreement, and how that relates to the correlation numbers you report.

The method does not deal with other constraints if subtitle translation, such as length limits

**Questions:**

You point out the contrast between chat and reasoning LLMs. But are you actually using reasoning LLMs to "think"? The prompt in the appendix asks just for a straight translation.

In Figure 4, you start with two differerent datasets - but are they actually different?

The reinforcement learning approach only tries to optimize vividness, or do I misunderstand something here?

There seem to be three variations in your preference optimization approach: (1) filtering out of small differences (Eqn2), importance scoring based on size of the set (Eqn3) and a dynamic beta. Which of these things make the method "adaptive" - I could not quite connect the name with the approach.

Do I read Table 4 right that your method wins against the human translation on all language pairs / metrics?

---

> ### Author Response · Authors · 2025-11-15
> **Reply to Reviewer i1xn**
>
> Thank you for your thoughtful review. Below we provide point-by-point responses to your comments and concerns.
>
> # W1: Lack of Inter-Annotator Agreement Check
>
> We conducted a human assessment consistency verification in Appendix B.2.1 (Figure 5).
>
> # W2: No Consideration of Subtitle Length Constraints
>
> In general, if subtitle translation process is followed by text-to-speech synthesis to generate target-language speech, length or duration constraints need to be considered. Controlling the output length is also a non-trivial problem. Since this paper mainly focuses on translation quality, we do not discuss this aspect.
>
> # Q1: Use of Reasoning LLMs for Actual Reasoning?
>
> The output of reasoning LLMs contains two parts: the “reason” and the “answer,” where the reasoning process is usually enclosed between the two tokens <think> and </think>. Although our prompt in the appendix asks for a straight translation, the reasoning LLM still first outputs the reason content and then the answer.
>
> # Q2: Clarification on Dataset Differences in Figure 4
>
> The datasets $\mathbb{D} _{\text{sft}}$ and $\mathbb{D} _{\text{alpo}}$ are two subsets obtained by randomly splitting dataset $\mathbb{D}$ at an 8:2 ratio, and they share the same distribution.
>
> # Q3: Scope of Reinforcement Learning Objective
>
> The main purpose of ALPO is to optimize vividness. LLMs already perform well on accuracy and naturalness, though ALPO also brings improvements on these two dimensions.
>
> # Q4: Meaning of “Adaptive” in ALPO Components
>
> In ALPO, all three components contribute to the “adaptive” nature of the method:
>
> - Filtering small differences (Eq. 2) adaptively excludes low-information lines, preventing noise during optimization.
>
> - Importance scoring (Eq. 3) adaptively reweights each line based on the diversity of sampled translations, allocating more learning signal to lines where vividness improvement is possible.
>
> - Dynamic $\beta$ adaptively scales the contrastive strength according to the actual reward gap of each segment.
>
> Overall, ALPO is designed to dynamically determine the contribution of each line during training based on its sampling diversity.
>
> # Q5: Interpretation of Human vs. Model Results in Table 4
>
> In machine translation tasks, it is normal that LLMs outperform humans. On one hand, the skill level of human translators varies; on the other hand, through large-scale training, LLMs have already acquired strong capabilities in translation accuracy. However, in the domain of subtitle translation, there is one aspect where LLMs are weaker than humans: leveraging video scene information during translation. For example, in the English-to-Chinese scenario, for the simple line “I’m off.”, an LLM relying solely on textual information would translate it as “我要走了” (“I gotta go.”). But in the actual scene, the character is speaking on the phone and intends to say he is about to hang up, so a human can correctly translate it as “我挂了” (“I will hang up.”). Therefore, in future subtitle translation research, we believe that multimodal translation capabilities should be further explored using MLLMs.

---

### Official Review · Reviewer_w6RR · 2025-10-31

**Soundness:** 2
**Presentation:** 2
**Contribution:** 2
**Rating:** 4
**Confidence:** 3

**Summary:**

The paper targets subtitle translation, arguing that good subtitles often require “liberal” renderings rather than literal word-for-word output. The authors propose a segment-level preference-optimization setup (ALPO): for each subtitle line, they sample multiple candidates, use an LLM-as-judge to pick local winners/losers, and train the model with a contrastive DPO-style loss; a few engineering choices (segment gating, prefix mixing, adaptive weighting) aim to stabilize learning and encourage fluency. Evaluation emphasizes three dimensions—accuracy, naturalness, and vividness—with both LLM-judged metrics and a small human study. Across several language directions, the ALPO-trained model improves over its SFT base and is competitive with strong chat/reasoning LLMs, particularly on lower-resource directions. Ablations suggest the local preference signal and the auxiliary knobs contribute to the gains.

**Strengths:**

The paper tackles a real-world application scenario—subtitle translation often benefits from more liberal renderings—and sets up a clean evaluation around that goal. The method is straightforward and well engineered: segment-level preference optimization with an LLM-as-judge, some human validation, and clear axes (accuracy/naturalness/vividness). The ablations are useful and suggest the components (e.g., segment gating, prefix mixing) actually matter. Empirically, the ALPO-trained model shows consistent gains over its SFT baseline and is competitive with strong chat/reasoning baselines, with gains on lower-resource directions.

**Weaknesses:**

Novelty and Missing references and baselines:

The work feels largely application-driven, and the methodological novelty is unclear beyond packaging known components (segment-level preference optimization with an LLM-as-judge) for subtitles. In particular, there’s no head-to-head comparison against fine-grained preference-learning baselines that operate below sequence level, and skips fine-grained baselines that are directly comparable to a segment-level method.

At minimum, it should include some of the following methods:

For fine-grained DPO:

(i) “Token-level Direct Preference Optimization” — Yongcheng Zeng et al., arXiv, 2024

(ii) “TIS-DPO: Token-level Importance Sampling for Direct Preference Optimization” — Aiwei Liu et al., ICLR, 2025

(iii) “SDPO: Segment-Level Direct Preference Optimization for Social Agents” — Aobo Kong et al., ACL (Long), 2025.

For MT specifically, a fine-grained SFT baseline like:

“Learning from Others’ Mistakes: Finetuning Machine Translation Models with Span-Level Error Annotations” — Lily H. Zhang et al., ICML, 2025,

should also be in scope. Adding these would test whether the gains come from segment-/token-level supervision itself rather than the particular packaging here.

**Questions:**

Fine-grained baselines. Why no head-to-head with fine-grained DPO (e.g., token- or segment-level) and span-level MT training (TWA)? Please add at least one strong token-level DPO baseline and a TWA-style span-supervised SFT baseline on the same data.

Credit assignment. How exactly are segment preferences aggregated across a subtitle to affect the full translation? Any cases where segment-level wins yield worse whole-sequence coherence? An ablation on seq- vs seg-only vs mixed would help.

Judge calibration. How sensitive are results to the choice/prompting of the LLM-as-judge? Please report inter-judge agreement, bias by style (literal vs liberal), and a calibration curve vs. human wins. Also, was any method used to overcome the LLM-as-judge biases (e.g. for label bias, ordering bias, etc.) or would that make a difference in the results.

Statistical rigor. For main tables, can you include confidence intervals and a simple permutation/bootstrap test for deltas? Also report the number of items per direction to assess power.

Gating and prefix-mixing. These look important. Could you share the exact gating rule, thresholds, and failure cases? What happens if you remove prefix mixing or replace it with random context shuffling?

Cost/efficiency. What is the end-to-end training and inference cost vs. your strongest baseline, and the quality-per-dollar or per-second trade-off?

Error analysis. Please include a brief breakdown of where ALPO wins/loses (e.g., idioms, wordplay, cultural references, timing constraints). This would make the “liberal translation” claim more concrete.

---

> ### Author Response · Authors · 2025-11-15
> **Reply to Reviewer w6RR**
>
> Thank you for your thoughtful review. Below are our point-by-point responses.
>
> # W1 & Q1: Novelty and Missing PO Baselines
>
> In fact, we conducted comparative experiments with other PO methods in Appendix C.4 (Table 13), and we also compared ALPO with different PO methods (including the SDPO method you mentioned) in Appendix C.5 (Table 14) under the social agent scenario to verify the applicability of ALPO to other tasks. The experiment in Table 13 was designed to investigate two questions:
>
> 1. whether vanilla DPO can directly address the goal of improving expressiveness and vividness in subtitle translation;
> 2. whether RL-based methods (e.g., PPO) can outperform direct optimization methods such as DPO.
>
> We present part of the results from Table 13 below, and we also include two token-level DPO-based methods you mentioned, TDPO and TIS-DPO (SDPO and TWA have not made their code open source):
>
> ||||en→zh|||zh→en||
> |:-:|:-:|:-:|:-:|:-:|:-:|:-:|:-:|
> |**Model**|**Train**|Acc|Nat|Vivi|Acc|Nat|Vivi|
> |Human|-|```83.6```|```82.9```|```71.6```|```82.9```|```80.2```|```73.2```|
> |DeepSeek-R1|In-context Learning|```90.4```|```85.6```|```70.7```|```88.5```|```85.6```|```73.6```|
> |Qwen2.5-14B|SFT|```86.5```|```82.1```|```59.2```|```85.2```|```80.1```|```54.9```|
> |Qwen2.5-14B|DPO|```87.1```|```82.2```|```63.3```|```86.1```|```81.1```|```60.2```|
> |Qwen2.5-14B|PPO|```88.3```|```83.2```|```70.1```|```85.9```|```82.3```|```70.1```|
> |Qwen2.5-14B|TDPO|87.4|82.7|64.0|86.0|81.3|61.2|
> |Qwen2.5-14B|TIS-DPO|88.5|83.1|65.9|86.2|83.3|62.4|
> |Qwen2.5-14B|ALPO (Ours)|```90.6```|```84.2```|```76.6```|```88.3```|```86.8```|```81.6```|
>
> We also conducted the experiment in Appendix C.3.1 (Table 12) to verify ALPO’s compatibility with different losses:
>
> ||||en→zh|||zh→en||
> |:-:|:-:|:-:|:-:|:-:|:-:|:-:|:-:|
> |**Method**|$\mathcal{L}_{\text{po}}$|Acc|Nat|Vivi|Acc|Nat|Vivi|
> |SFT|-|```86.5```|```82.1```|```59.2```|```85.2```|```80.1```|```54.9```|
> |ALPO|DPO|```90.6```|```84.2```|```76.6```|```88.3```|```86.8```|```81.6```|
> |ALPO|GRPO|```91.2```|```85.0```|```77.1```|```87.2```|```84.9```|```81.3```|
> |ALPO|TDPO|89.9|84.1|75.4|87.4|85.3| 80.4|
> |ALPO|TIS-DPO|91.0|84.3|76.9|88.5|86.1|81.8|
>
> (This ```format``` indicates results reported in our paper.)
>
> The results in the first table show that TDPO and TIS-DPO fall behind PPO and ALPO. In the second table, when applying TDPO and TIS-DPO losses within the ALPO framework, their performance improves significantly.
>
> The fundamental reason is whether the method tackles the core challenge of this study. The main challenge is that the LLM response contains multiple segments, and the reward signal acts on each segment independently, meaning that each segment has its own reward signal (Line 326–327). In Appendix D, we explain why this challenge leads to degraded performance for outcome-supervised methods such as DPO. Specifically, if the LLM output $y$ consists of multiple segments, i.e., $y=(y_1,\cdots,y_n)$ and $\pi_{\theta}(y|x)=\prod_{i=1}^{n}\pi_{\theta}(y_{i}|x,y_1,\cdots ,y_{i-1})$, then the following DPO loss term becomes:
>
> $$\text{log}\frac{\pi _{\theta }(y^w|x)}{\pi _{\text{ref}}(y^w|x)}-\text{log}\frac{\pi _{\theta }(y^l|x)}{\pi _{\text{ref}}(y^l|x)}=\sum _{i=1}^{n}\left [\text{log}\frac{\pi _{\theta }(y_i^w|\textcolor{red}{x,y _{1:i-1}^w})}{\pi _{\text{ref}}(y_i^w|\textcolor{red}{x,y _{1:i-1}^w})}-\text{log}\frac{\pi _{\theta }(y_i^l|\textcolor{red}{x,y _{1:i-1}^l})}{\pi _{\text{ref}}(y_i^l|\textcolor{red}{x,y _{1:i-1}^l})}\right ]$$
>
> That is, when using vanilla DPO (an outcome-supervised method), the preferred output $y_i^w$ and dispreferred output $y_i^l$ for each segment are conditioned on *different prefixes*. This mismatch hinders the contrastive optimization. The above baselines are not designed to resolve this issue, whereas ALPO is explicitly designed to do so. Its effectiveness arises from two key designs:
>
> 1. unifying the prefix condition for preferred and dispreferred outputs of the same segment as $p_i$;
> 2. using the gating function $\mathbf{1}(s_{i})$ to filter out low-diversity noisy segments.
>
> # Q2: Credit Assignment
>
> We evaluated the impact of coarse-grained vs. fine-grained sampling in Table 13. The results show that fine-grained sampling yields clear performance improvements.
>
> # Q3: Judge Calibration
>
> In Table 3, the robustness of LLM-as-a-Judge and mitigation of model bias is ensured not by using multiple evaluations from the same model, but by using multiple models (DeepSeek-V3.1, Claude Sonnet 4, and GPT-5). See results for individual models in Table 10.
>
> # Q4 & Q6: Statistical Rigor and Cost/Efficiency
>
> We appreciate the suggestion regarding statistical rigor. Additional details on the experimental setup, including computational resources and test set size, are provided in Appendix B.1.

---

> > ### Comment · Reviewer_w6RR · 2025-11-28
> >
> > Follow-up questions:
> >
> > Q1: When running the vanilla baselines TDPO and TIS-DPO, how did you choose the pairs (preferred and dispreferred) for those baselines? Did you choose the same pairs as ALPO? How can we disentangle whether the improvement comes from your proposed loss function or simply from the data construction strategy (i.e. construction of pairs)?
> >
> > Q2: In Section 3.2, you rely on the assumption that lower back-translation BLEU scores indicate a higher degree of "liberal" translation. While you show this trends across domains (e.g., Literature vs. Law), is there any quantitative evidence—such as correlation with human ratings of "vividness" or "liberalness"—that validates this metric specifically for subtitle translation? Without this, it is difficult to distinguish whether a low BLEU score indicates a desirable "liberal" translation or simply a semantic drift.
> >
> > Q3: I noticed that while you added TDPO and TIS-DPO, the response did not address the comparison with TWA ("Learning from Others’ Mistakes", Zhang et al., 2025). Given that TWA also operates on span-level error corrections (similar to the segment level in ALPO) while does not suffer from the token-drift issue of TDPO and TIS-DPO, it appears to be a more direct competitor to ALPO than the token-level baselines provided. Could you clarify why this comparison was omitted?
> >
> > A broader issue is the positioning of the paper. A significant portion of the work focuses on the heuristics of the specific application (subtitle translation) rather than the generalizable benefits of the segment-level DPO method (ALPO). If ALPO is the primary contribution, the paper would benefit from disentangling the method from the specific engineering choices of the use case. Currently, it is difficult to assess if ALPO is a general-purpose improvement for long-form generation or a heuristic specifically tuned for subtitles.

---

> ### Author Response · Authors · 2025-11-17
> **Reply to Reviewer w6RR (Continued)**
>
> # Q5: Gating and Prefix-mixing
>
> The gating rule is: when $|\mathcal{T} _{i}|\leq 3$ or $\max(\mathcal{E} _{i})-\min(\mathcal{E} _{i})\leq 5$, we set $\mathbf{1}(s _{i})=0$ (Line 333–334). The ablation study for gating and prefix-mixing is shown in Table 5.
>
> # Q7: Error Analysis
>
> We present a case study and analysis in Appendix C.2. In subtitle translation, there is one aspect where LLMs are weaker than humans: the ability to translate by incorporating *scene* and *plot* information from the video. For example, in the English-to-Chinese setting, a simple line like “I'm off.” would be translated by an LLM (based solely on text) into “我要走了” (meaning “I gotta go.” in English). But in the actual scene, the character saying this line is on the phone and intends to express that he is about to hang up, so a human would correctly translate it as “我挂了” (“I will hang up.”). Another example from Appendix C.2 is the sentence “Nature has made us intolerant to change”. For the word “Nature”, LLMs tend to translate it as “自然” (“natural world”), while humans, based on the plot context, can correctly translate it as “天性” (“innate disposition”). These two examples illustrate that even if we provide LLMs with the surrounding subtitle context, they may still mistranslate, because the key information required comes from other modalities. This reflects another particularity of subtitle translation: subtitles are texts that are tightly coupled with the video and audio modalities. Therefore, in future subtitle translation research, we believe it is necessary to leverage MLLMs to further explore multimodal translation capabilities.

---

> ### Author Response · Authors · 2025-11-27
> **Gentle Reminder regarding our response**
>
> Dear Reviewer w6RR,
>
> I hope this message finds you well.
>
> As the discussion period is entering its final week, we wanted to ensure that our previous responses and revisions have satisfactorily addressed your concerns.
>
> If there are any additional points or feedback you would like us to consider, or if any clarifications are still needed, please let us know. Your insights are invaluable to us, and we are eager to address any remaining issues to further improve our work before the deadline.
>
> Thank you strictly for your time and effort in reviewing our paper.
>
> Best regards, The Authors

---

> ### Author Response · Authors · 2025-11-28
> **Further Reply to Reviewer w6RR**
>
> Thank you for your response. We clarify the remaining questions below.
>
> # Q1: Detailed Experimental Setup
>
> We used **the same pairs as ALPO** when running TDPO and TIS-DPO. These preferred and dispreferred pairs originate from the sampling data checkpoints saved during our execution of ALPO; we did not perform a separate sampling inference process for TDPO and TIS-DPO. Moreover, the selection criteria for preferred and dispreferred samples are exactly consistent with ALPO, i.e., "random from top-3 as preferred, third-lowest as dispreferred" (Lines 345-347). Additionally, given the distinct performance gap between TDPO/TIS-DPO and PPO/ALPO (e.g., in the Vividness dimension, TIS-DPO: 62.4, PPO: 70.1, ALPO: 81.6), such a significant performance difference is unlikely to be caused by the *data construction strategy*.
>
> We show zh→en case studies illustrating performance differences across models. High-quality translations are marked in this ```format```:
>
> ||Human|SFT|TIS-DPO|GPT-5|ALPO|
> |:-|:-|:-|:-|:-|:-|
> |现在恐怕还不是谈情说爱的时候|I’m afraid this isn’t the time for you two to be ```lovey-dovey```.|I'm afraid now is not the time to talk about love.|I'm afraid now is not the time for romance.|It’s probably not the time yet to talk about love.|I’m afraid it’s not the right time for ```lovey-dovey``` stuff just yet.|
> |只可惜这和煦的暖阳以后再也看不到了|It’s a pity that I won’t be able to feel the warm sunshine anymore.|It's a pity that I won't be able to see the warm sunshine again.|The only pity is that this ```genial``` sunshine will never be seen hereafter.|It’s just a pity this warm, ```gentle``` sunlight--I’ll never see it again.|What a shame, I’ll never get to ```bask in``` the warm sunshine ever again...|
> |青山常在 绿水长流|The world will ```remain the same```.|Green hills always remain; green waters long flow.|The green mountains are always there; the green water flows forever.|The green hills ```endure```, the blue rivers flow.|May the mountains stand tall and the waters flow on.|
> |她一定有三头六臂|I’m sure she has many arms and legs.|She surely has three heads and six arms.|She must have three heads and six arms.|She must have three heads and six arms or something.|She must be a ```superwoman```.|
>
> # Q2: Unconvincing BLEU Metric
>
> The initial motivation for the investigation in Section 3.2 was "hypothesis-then-verification." Specifically, we posit that "domains like Literature tend more towards liberal translation, while domains like Legislation tend more towards literal translation." This is intuitively perceived by many, and we aimed to verify this quantitatively (Lines 159-161, Line 177).  Therefore, we selected BLEU and ChrF++ for verification. The underlying rationale is that if a translation is more liberal, the surface-level matching between the back-translated text and the source text will be lower. Moreover, the experiments in Table 1 aligned with our expectations, showing distinct differences between literal and liberal domains (maximum difference: BLEU > 10, ChrF++ > 20). In summary, BLEU serves as a heuristic for style verification, whereas in the experiments of Section 5, vividness/liberalness is verified via LLM-as-a-Judge.
>
> Below we show several case studies to intuitively demonstrate the effectiveness of the BLEU metric. Text matching the source is marked in this ```format```:
>
> |Source|Translation|Back-Translation|BLEU|
> |:-|:-|:-|:-|
> |现在恐怕还不是谈情说爱的时候|I'm afraid now is not the time to talk about love. (literal)|```恐怕现在不是谈```爱情```的时候```|15.79|
> |现在恐怕还不是谈情说爱的时候|I’m afraid this isn’t the time for you two to be lovey-dovey. (liberal)|```恐怕现在不是```你们俩卿卿我我```的时候```|10.07|
> |
> |她一定有三头六臂|She must have three heads and six arms. (literal)|```她一定有三```个```头```和```六```条胳膊|34.48|
> |她一定有三头六臂|She must be a Superwoman. (liberal)|```她一定```是个女超人|13.75|
>
> We also discussed this issue with Reviewer VFQE. Please refer to "Second Point in W2" in our Rebuttal with him.
>
> # Q3: Lack of Comparison with TWA
>
> TWA has not open-sourced their code. We apologize for forgetting to clarify this in our first rebuttal.
>
> In fact, span-level is a granularity between token/word-level and segment/sentence-level, i.e., token/word (TDPO/TIS-DPO) < span (TWA) < segment/sentence (ALPO) < outcome (DPO). We believe it leans more towards Token/Word, or more accurately, fine-grained levels. Therefore, experiments on TDPO/TIS-DPO demonstrate the limitations of token/word-level methods on the subtitle translation task. The core reason, as verified by our experiments and theory (Appendix C.4 and D), is that *the reward signal acts on each segment independently*, rather than the issue of granularity.
>
> # Broader Issue: Generalization of ALPO
>
> In Appendix C.5 (Table 14), we specifically verified the applicability of ALPO in the **multi-turn social agent** scenario (long-form generation task). We compared it with SDPO (code not open-sourced), a method designed specifically for this task, and achieved comparable performance.
>
> Please feel free to let us know if you have any further questions.

---

### Official Review · Reviewer_VFQE · 2025-11-01

**Soundness:** 2
**Presentation:** 3
**Contribution:** 3
**Rating:** 6
**Confidence:** 4

**Summary:**

This manuscirpt focus on the problem of subtitle translation using LLM. Subtitle translation refers to the machine translation task in the scenarios such as visual media and movie (visual information is not used), where the emotion and atomosphere should be considered. The authors first investigate the subtitle translation by examine exisiting LLMs with comparing the literal and liberal translation in different domain. Then, the authors proposed ALPO, a preference optimization framework that considers the contexts and segmentation specifically needed for subtitle translation. The effectiveness of the method is evaluated by LLM-as-a-judge and human measurement on unclear number of examples.

**Strengths:**

1. The manuscript is well-written and I could follow the logic and understand the background.
2. Comprehensive benchmarks and exisiting LLMs are evaluated in the experiments.
3. The problem of subtitle translation (maybe in a more general sense, machine translation in diverse tones and scenarios), is an important and emergent problem as SOTA LLMs has now reached a very strong performance on literal translation.

**Weaknesses:**

1. Although the topic is interesting and problem is important, the evaluation method is a bit weak and lack of soundness. See my questions for specific points below.
2. The technical challenges of subtile translation are context-dependency and segmentation, etc., which are not discussed in depth. Intead, the point of liberal translation that the authors highlighted is vague in this manuscirpt: for example, there is a gap between lower BLEU and hihgher liberal translation, this could be simply caused by mis-translation from the models. More justification would be helpful in Section 3.2.
3. In Section 5, the experiments use LLM-as-a-judge to evaluate the models assuming that LLM has already understand the "emotion" and "atomospeher", then why those LLM still need optimization for those two aspects? Although the BLEU correlation is shown in Section 3 but it's not directly related to the scope in vividness evaluation.


Minor comments:
1. In Figure3 caption: DS-R1 is the abbr. for Deepseek-R1 can be stated, though most people could guess the meaning. In addtion, chat-models and reasoning-models can be somehow visually diffrentiated, as you're comparing the patterns for those two groups.
2. A previous work about word-level preference reward for machine translation [1], could be related this work as subtitle translation needs sentence-level context.

[1] Word Alignment as Preference for Machine Translation.EMNLP 2024.

**Questions:**

1. Can I understand ALPO as a context-aware sampling (step 2) + DPO (step 3)?
2.This is related to limitation 3, is it possible to add an ablation experiments that expilictly asking reasoning LLM to consider vividness, emotion and atomospher withouth ALPO optimization?

---

> ### Author Response · Authors · 2025-11-15
> **Reply to Reviewer VFQE**
>
> Thank you for your thoughtful review. Next, I will provide detailed explanations regarding your concerns.
>
> # First Point in W2: Underanalyzed Task Challenges
>
> Context-dependency and segmentation are indeed important challenges in subtitle translation. Our approach addresses this issue by grouping the lines of a program and including a group of 25 lines within each prompt. Additionally, each group overlaps with its preceding and following groups by 5 lines, meaning that one prompt actually contains 35 lines. In this way, we ensure that each line has access to sufficient contextual information during translation. However, these details belong to the data preprocessing stage, so we did not discuss them extensively in the paper and instead expect readers to find them in our released code. Beyond this, the core goal of this paper is to explore how to improve the expressiveness and vividness of LLMs for subtitle translation. At the technical level, the primary challenge lies in the fact that an LLM response contains multiple segments, and the reward signal acts on each segment individually, meaning that each segment receives its own reward signal. The full design of ALPO is dedicated to addressing this goal and challenge.
>
> # Second Point in W2: Unconvincing BLEU Metric
>
> Domains such as literature and subtitle translation favor liberal translation, while more serious domains such as law and medicine favor literal translation. This is a straightforward and intuitive common-sense perspective. In this paper, we aim to quantify this intuition to illustrate our motivation. Additionally, we evaluate the degree of liberal and literal translation exhibited by chat and reason models to examine the vividness translation capability of today’s LLMs in the subtitle translation task.
>
> In conducting this empirical investigation, we use BLEU, a traditional translation evaluation metric, to measure the degree of liberal translation. This could indeed be confusing for researchers in traditional machine translation, where *low BLEU = mistranslation or low translation quality*. However, fundamentally, BLEU is an n-gram based metric that measures surface-level matching between texts. In Sec. 3.2, we use BLEU to evaluate the matching between LLM back-translations and human references, and in Sec. 3.3 we compare the matching between translations produced by different LLMs. It is worth noting that both the human references and the LLM-generated translations are of relatively high quality and are unlikely to contain mistranslations. Therefore, we believe that BLEU in this paper reflects the degree of liberal translation. More broadly, we argue that in the LLM era, machine translation research has shifted from improving translation accuracy toward adapting to the specific requirements of different domains and scenarios.
>
> It is also worth highlighting that the observations from Sec. 3.3 are consistent with the results in Table 3. Sec. 3.3 shows that chat models tend to prefer literal translation, which corresponds to their relatively higher accuracy and lower vividness in Table 3. Conversely, reason models are capable of performing liberal translation and therefore achieve relatively higher vividness and lower accuracy in Table 3.
>
> We will strengthen the discussion of these points in the final version of the paper.
>
> # W3 & Q1: The Necessity of Optimization
>
> In fact, across many domains and tasks involving LLMs, there is an important empirical observation: for many tasks, LLMs tend to be better evaluators / critics / judges than performers / generators / actors. For example, in math reasoning, LLMs are stronger as math-solution critics than solvers. Given a chain of mathematical reasoning, an LLM can often identify whether the logic holds, where the mistake lies, and whether the final answer is reasonable. However, when the LLM solves math problems itself, its error rate is high. Similarly, in factual consistency / hallucination detection, LLM judges can sometimes outperform human annotators in determining consistency between claims and evidence, but the same LLM may still hallucinate when answering factual questions directly.
>
> Similarly, in our task, we explicitly ask chat/reason models in the prompt to use in-context learning to produce expressive and vivid translations. However, the results in Table 3 show that chat models do not effectively follow this instruction, and reason models rarely reach human-level performance. Below we show a portion of the full prompt (the complete version can be found in Table 15):
>
> > The English translation must be expressive and vivid, effectively conveying the atmosphere, emotions, and tone of the original Chinese lines.
>
> These observations demonstrate that although LLMs can evaluate vividness well, they cannot reliably produce vivid translations, which makes optimization necessary.
>
> # Minor Comments
>
> Thank you for your suggestions regarding our paper writing. We will revise the manuscript accordingly.

---

> > ### Comment · Reviewer_VFQE · 2025-11-27
> >
> > Thanks for the detaailed responses.
> >
> > 1. I have to clarify that I didn't mean that low BLUE = mis-translation. However, what I wanted to ask is: BLUE, as a surface-level metric, does not strictly correlate to vividness, which is not well-defined in this work.
> >
> > 2. I appreciate the authors' explanation regarding the background of LLM-as-a-judge. I do agree with thats LLM can work better  on judges/evaluators that problem solver in many tasks. However, the reason of this is natural: in many problem settings the ground-truth answer is well defined and reference-based, the evaluation tasks has been simplified to judge the consistency between model output and groundtruth. In this work, the vividness is an abstract concept which is way harder to define. That's why I suggested that a deeper study on this issue will be very valuable.
> >
> > I will keep my rating.

---

> ### Author Response · Authors · 2025-11-27
> **Further Reply to Reviewer VFQE**
>
> Thank you for your response, which has helped clarify your concerns. We would like to make the following further clarifications:
>
> 1. The initial motivation for the investigation in Section 3.2 was "hypothesis-then-verification." Specifically, the intuition that domains such as Literature tend toward liberal translation, while domains like Legislation tend toward literal translation, is widely shared. We aimed to quantitatively verify this intuition (Lines 159-161, Line 177). Therefore, we selected metrics such as BLEU and ChrF++ for verification. The underlying rationale is that if a translation is more liberal, the surface-level matching between the back-translated text and the source text will be lower. Moreover, the experiments in Table 1 aligned with our expectations, showing distinct differences between literal and liberal domains (maximum difference: BLEU > 10, ChrF++ > 20). Based on this, we used the BLEU metric in Section 3.3 to measure the liberal translation performance of chat and reason models, and the experiments in Table 3 further validated the results of Section 3.3.
>
> Below we present several case studies to intuitively demonstrate the effectiveness of the BLEU metric. Text matching the source is marked in this ```format```:
>
> | Source | Translation | Back-Translation | BLEU |
> |:-|:-|:-|:-|
> | 现在恐怕还不是谈情说爱的时候 | I'm afraid now is not the time to talk about love. (literal translation) | ```恐怕现在不是谈```爱情```的时候``` | 15.79 |
> | 现在恐怕还不是谈情说爱的时候 | I’m afraid this isn’t the time for you two to be lovey-dovey. (liberal translation) | ```恐怕现在不是```你们俩卿卿我我```的时候``` | 10.07 |
>
> | Source | Translation | Back-Translation | BLEU |
> |:-|:-|:-|:-|
> | 她一定有三头六臂 | She must have three heads and six arms. (literal translation) | ```她一定有三```个```头```和```六```条胳膊 | 34.48 |
> | 她一定有三头六臂 | She must be a Superwoman.  (liberal translation) | ```她一定```是个女超人 | 13.75 |
>
> 2. We agree on the importance of investigating the ability of LLMs to evaluate and perform on certain abstract concepts or dimensions. Validating this capability across different tasks can sometimes be challenging. In translation scenarios, the evaluation of abstract dimensions such as expressiveness, vividness, and colloquialism often varies due to human subjectivity and is influenced by linguistic and cultural biases; therefore, in-depth research in this area is necessary. In this paper, we validated the LLM's ability to evaluate vividness by verifying its alignment with human evaluation, which we believe is a credible approach. We hope that the research in this paper can offer insights for other similar scenarios and tasks.
>
> Please feel free to let us know if you have any further questions or concerns.

---

### Official Review · Reviewer_Wkwm · 2025-11-05

**Soundness:** 2
**Presentation:** 3
**Contribution:** 2
**Rating:** 4
**Confidence:** 4

**Summary:**

This paper looks at translating subtitles for visual media. They introduce a novel Preference Optimization method (ALPO) that looks at local outputs instead of the full ones. Overall, the paper covers a lot of things in depth, but I have two main questions (in question section below) that I would like to see answered that will really impact my review scores.

**Strengths:**

This paper has a lot of positive contributions. The analysis of domains on literal vs liberal translations is quite nice. The numerous experiments on recent models encompassing a lot of modern SOTA LLMs. A potentially interesting new PO algorithm. Empirically showing Table 3 that we cannot always beat humans (which makes sense since they likely use more modalities to translate as well). A human evaluation in section 5.3

**Weaknesses:**

The proposed method is a novel PO algorithm, but the comparisons are only to other models, not other PO methods. The authors claim that local is better, and thus propose ALPO as opposed to using full sequence outputs such as DPO. However, this is never empirically shown (as far as I can tell).

Differentiating the subtitle translation task from OpenSubtitles (Tiedemann 2016; Lison and Tiedemann 2016). It is mentioned as a core claim, but is only mentioned in section B.2.2

**Questions:**

Normally, PO tasks are done on full sequence outputs which is often touted as a benefit. I guess you could have preferences for partial outputs and that is not a problem, but it seems that in many ways, that is more similar to SFT with a slightly different loss - since you just update at every token autoregressively. Am I understanding this correctly? Regardless, this is a semantic definition that really does not impact the paper too much. My main question is how would a sequence level method (DPO, CPO, KTO, SimPO, etc. - choose one) compare to your method? Essentially, I am asking for another row in Table 3 with one of these.

Your first claim at the end of your introduction states "We introduce the visual media subtitle translation task for the first time..." It isn't clear to me that this is the first time subtitle translation is done. You even cite OpenSubtitles in B.2.2. How is your claim here different that OpenSubtitles? Is this just a new dataset? If so, that is fine, but the claim should be about that - and not that this is a novel task. Or am I missing something key to your task that is new?

---

> ### Author Response · Authors · 2025-11-15
> **Reply to Reviewer Wkwm**
>
> Thank you for your thoughtful review. Below we provide responses to your concerns.
>
> # W1 & Q1: Lack of Comparison with Other PO Methods
>
> In fact, we conducted the comparison experiments between ALPO and other PO methods in Appendix C.4 (Table 13). We also compared ALPO with different PO methods in Appendix C.5 (Table 14) under the social agent scenario to verify the applicability of ALPO to other application tasks. Appendix C.4 was designed to investigate two questions:
>
> 1. whether DPO can directly achieve the goal of improving expressiveness and vividness in subtitle translation;
> 2. whether RL methods (such as PPO) can outperform direct optimization methods such as DPO.
>
> We present part of the results from Table 13 below, and following reviewer w6RR, we also test two token-level PO methods, TDPO and TIS-DPO:
>
> ||||en→zh|||zh→en||
> |:-:|:-:|:-:|:-:|:-:|:-:|:-:|:-:|
> |**Model**|**Train**|Acc|Nat|Vivi|Acc|Nat|Vivi|
> |Human|-|```83.6```|```82.9```|```71.6```|```82.9```|```80.2```|```73.2```|
> |DeepSeek-R1|In-context Learning|```90.4```|```85.6```|```70.7```|```88.5```|```85.6```|```73.6```|
> |Qwen2.5-14B|SFT|```86.5```|```82.1```|```59.2```|```85.2```|```80.1```|```54.9```|
> |Qwen2.5-14B|DPO|```87.1```|```82.2```|```63.3```|```86.1```|```81.1```|```60.2```|
> |Qwen2.5-14B|PPO|```88.3```|```83.2```|```70.1```|```85.9```|```82.3```|```70.1```|
> |Qwen2.5-14B|TDPO|87.4|82.7|64.0|86.0|81.3|61.2|
> |Qwen2.5-14B|TIS-DPO|88.5|83.1|65.9|86.2|83.3|62.4|
> |Qwen2.5-14B|ALPO (Ours)|```90.6```|```84.2```|```76.6```|```88.3```|```86.8```|```81.6```|
>
> (This ```format``` indicates results reported in our paper.)
>
> ALPO outperforms all PO baselines. It is important to emphasize that the main challenge of this study is that the LLM’s response contains multiple segments, and the reward signal acts on each segment independently, meaning that each segment has its own reward signal (Line 326–327). ALPO achieves its strong performance precisely because it solves the core challenge of local reward supervision. In Appendix D, we explain why this challenge causes outcome-supervised methods such as DPO to perform poorly. Specifically, if the LLM output $y$ consists of multiple segments, i.e., $y = (y_1, \cdots, y_n)$, and satisfies $\pi_{\theta}(y\mid x)=\prod_{i=1}^n\pi_{\theta}(y_i\mid x, y_1,\cdots, y_{i-1})$, then the following item in the DPO loss becomes:
>
> $$\text{log}\frac{\pi _{\theta }(y ^w \mid x)}{\pi _{\text{ref}}(y ^w \mid x)}-\text{log}\frac{\pi _{\theta }(y ^l \mid x)}{\pi _{\text{ref}}(y ^l \mid x)}=\sum _{i=1}^{n}\left [\text{log}\frac{\pi _{\theta }(y _i^w \mid \textcolor{red}{x,y _{1:i-1}^w})}{\pi _{\text{ref}}(y _i^w \mid \textcolor{red}{x,y _{1:i-1}^w})}-\text{log}\frac{\pi _{\theta }(y _i^l \mid \textcolor{red}{x,y _{1:i-1}^l})}{\pi _{\text{ref}}(y_i^l \mid \textcolor{red}{x,y _{1:i-1}^l})}\right ]$$
>
> That is, when using outcome-supervised methods such as DPO, during the optimization of each segment, the preferred output $y_i^w$ and the dispreferred output $y_i^l$ are conditioned on different prefixes. This mismatch hinders the contrastive optimization. The above baselines are not designed to address this issue, whereas ALPO is specifically designed for this purpose. Its effectiveness comes from two key designs:
>
> 1. unifying the prefix conditions of the preferred and dispreferred outputs of the same segment to $p_i$;
> 2. using a gating function $\mathbf{1}(s_i)$ to exclude low-diversity noisy segments from the optimization process.
>
> # W2 & Q2: Difference from OpenSubtitles
>
> Subtitle translation is typically a research task within the humanities and social sciences (such as Translation Studies, Literature, and Communication Studies). This paper is the first to computationalize this domain, thereby transforming it into a computational task within the field of Computer Science.
>
> In fact, OpenSubtitles [1] merely released a subtitle parallel corpus. OpenSubtitles does not define a task, nor does it introduce the special objectives, stylistic requirements, preference structures, or evaluation dimensions behind subtitle translation. Our work is not simply “collecting yet another dataset.” We are the first to introduce “subtitle translation” as an independent task with clear task definition, evaluation criteria, and methodological framework, and to conduct systematic study of it. In short:
>
> - OpenSubtitles = new corpus
> - Our work = task definition + translation objective + evaluation framework + optimization method + new corpus
>
> Another related work is VideoDubber [2] (which we acknowledge in Line 370), but it only focuses on speech duration consistency, does not address translation quality and define the task.
>
> In conclusion, to the best of our knowledge, no prior work studies subtitle translation as a computational translation task.
>
> >[1] OpenSubtitles2016: Extracting large parallel corpora from movie and TV subtitles, LREC 2016 \
> [2] Videodubber: Machine translation with speech-aware length control for video dubbing, AAAI 2023

---

> ### Author Response · Authors · 2025-11-27
> **Gentle Reminder: Feedback on Response**
>
> Dear Reviewer Wkwm,
>
> I hope this message finds you well. As the discussion period is entering its final week, we wanted to ensure we have addressed all your concerns satisfactorily, particularly regarding the comparison with other PO methods and the potential overclaims relative to OpenSubtitles.
>
> If there are any additional points or feedback regarding our new results and explanations that you'd like us to consider, please let us know. Your insights are invaluable to us, and we are eager to engage in further discussion to resolve these issues.
>
> Thank you for your time and effort in reviewing our paper.
>
> Best regards, The Authors

---

### Author Response · Authors · 2025-11-26
**General Response**

We sincerely thank all reviewers for their time and constructive feedback. We are encouraged by the positive comments regarding the novelty and effectiveness of our work. Below, we summarize the consensus on the strengths and concerns raised across the reviews. We also provide brief responses to the primary weaknesses and outline the corresponding revisions we commit to incorporating into the final version of the paper.

# Strengths

1. The proposed task and method exhibit novelty ($\textcolor{red}{\text{Wkwm}}$, $\textcolor{green}{\text{VFQE}}$, $\textcolor{magenta}{\text{i1xn}}$)

2. This paper is well-written and easy to understand and follow ($\textcolor{green}{\text{VFQE}}$, $\textcolor{magenta}{\text{i1xn}}$)

3. The empirical analysis of literal vs. liberal translation is in-depth and insightful ($\textcolor{red}{\text{Wkwm}}$, $\textcolor{magenta}{\text{i1xn}}$)

4. The proposed ALPO method is well-designed and addresses the main challenges of this task ($\textcolor{red}{\text{Wkwm}}$, $\textcolor{blue}{\text{w6RR}}$, $\textcolor{magenta}{\text{i1xn}}$)

5. Extensive experiments and ablations verify the effectiveness of ALPO ($\textcolor{red}{\text{Wkwm}}$, $\textcolor{green}{\text{VFQE}}$,  $\textcolor{blue}{\text{w6RR}}$)

6. Other strengths: Meaningful human evaluation ($\textcolor{red}{\text{Wkwm}}$, $\textcolor{blue}{\text{w6RR}}$), inspiring for other machine translation scenarios ($\textcolor{green}{\text{VFQE}}$), reasonable evaluation framework ($\textcolor{blue}{\text{w6RR}}$)

# Weaknesses

1. Lack of Comparison with Other PO Methods（$\textcolor{red}{\text{Wkwm}}$, $\textcolor{blue}{\text{w6RR}}$）

> Author response: We have already conducted comparative experiments with vanilla DPO and PPO in Appendix C.4 (Table 13), and we have added results for two token-level DPO-based methods, TDPO and TIS-DPO.

2. Concern regarding the Overclaim of "introduce the task for the first time": Difference from OpenSubtitles（$\textcolor{red}{\text{Wkwm}}$）

> Author response: Subtitle translation is typically a research task within the humanities and social sciences. This paper is the first to computationalize this domain, thereby transforming it into a computational task within the field of Computer Science. The OpenSubtitles paper merely released a new dataset, whereas our paper covers the task definition, translation objective, evaluation framework, optimization method, and a new corpus.

3. Unconvincing BLEU Metric（$\textcolor{green}{\text{VFQE}}$）

> Author response: Section 3.2 follows a 'hypothesis-then-verification' framework. Based on the intuition that liberal translation levels differ by field, we utilized quantitative metrics to verify this hypothesis. Human and LLM translations are less prone to mistranslation, so a low BLEU of back-translation more likely indicates a higher degree of liberal translation.

4. The Necessity of Training Optimization（$\textcolor{green}{\text{VFQE}}$）

> Author response: Various domains have verified an important empirical observation: for many tasks (e.g., math reasoning, hallucination detection), LLMs tend to be better evaluators / critics / judges than performers / generators / actors.

5. Lack of Inter-Annotator Agreement Check（$\textcolor{magenta}{\text{i1xn}}$）

> Author response: We have already conducted a human assessment consistency verification in Appendix B.2.1 (Figure 5).

6. No Consideration of Subtitle Length Constraints（$\textcolor{magenta}{\text{i1xn}}$）

> Author response: In this paper, we primarily focus on the quality of subtitle translation; the length/duration control of the translation is also a topic worth dedicated research.

# Revisions

1. Move the experiments in Appendix C.4 (Table 13) to the main text and add the results of TDPO and TIS-DPO.

2. Writing modifications: Add the references mentioned by the reviewers, improve the presentation of Figure 3 and modify several claims to avoid misunderstanding.

---

### Meta-Review · Area_Chair_BHde · 2026-01-03

**Summary:**

This paper focused on the specialized translation task, especially under visual media subtitle scenario. The key challenges are the gap between general domains and vertical domains, weak expressiveness and vividness in subtitle translation, and fine-grained local preference alignment. This work employed LLM-as-a-Judge and preference optimization techniques to build a customized subtitle translation LLM, revealing interesting findings such as the favor of subtitle translation and more literal translations by chat LLMs. This work further proposed a preference alignment strategy Adaptive Local Preference Optimization (ALPO) for fine-grained local preference alignment.

The main strengths are (1) the research problem and challenges of vertical domain translation is important and realistic, (2) the evaluation and empirical findings are insightful, (3) the performance gain over SFT and baselines are consistent. The major weaknesses and concerns are (1) the overclaim of "introduce the task for the first time" as well as the difference between related works such as OpenSubtitles, where the authors clarified the differences in the rebuttal but the importance and overall novelty are not explained very well, (2) missing comparisons with token-level PO methods which are further provided in the rebuttal.

The overall quality of this paper is good, considering the insights from the empirical investigation of LLMs for subtitle translation, the consistent performance gain, and the contributions to a specific vertical domain translation problem. The major weaknesses are the scope of the paper, the general contributions to the community, and the potential overclaim of the novelty. The final recommendation is borderline accept.

**Reviewer Concerns:**

Most of the concerns are well addressed, but the clarification of the novelty and contributions corresponding to the "first time" claim are not very clear. Please see "Reviewer Scores" for more details.

**Reviewer Scores:**

Reviewer w6RR with score 4 has engaged in the discussion and asked three more questions. Area Chair thinks these three questions were addressed by authors' follow-up answers. For the border issue that the main focus of this paper is domain-specific task rather than general approach. Area Chair thinks it is okay for a specific approach to a specific problem, as the paper also provided empirical insights from the investigation section. Also, the proposed approach is applicable to other application task called social language agent multi-turn interaction, although one additional task is not general enough. Even if Reviewer w6RR would not increase their score, the weaknesses are not strong concerns to reject the paper but make the paper borderline.

Reviewer Wkwm with score 4 requested the clarification of the novelty and contributions. The authors explained that the "first" means "transforming it into a computational task". However, it is not clear how novel such transformation is and how important the transformation is, which hinders the novelty of this work. More clarifications are needed to emphasis the contributions. The score might not be changed.

---

### Decision · Program_Chairs · 2026-01-26

Accept (Poster)